# Cell size sensing in animal cells coordinates anabolic growth rates and cell cycle progression to maintain cell size uniformity

Miriam Bracha Ginzberg[1,2], Nancy Chang[2], Heather D'Souza[2], Nish Patel[2], Ran Kafri[2]*, Marc W Kirschner[1]*

[1]Department of Systems Biology, Harvard Medical School, Boston, United States; [2]Cell Biology Program, The Hospital for Sick Children, Toronto, Canada

**Abstract** Cell size uniformity in healthy tissues suggests that control mechanisms might coordinate cell growth and division. We derived a method to assay whether cellular growth rates depend on cell size, by monitoring how variance in size changes as cells grow. Our data revealed that, twice during the cell cycle, growth rates are selectively increased in small cells and reduced in large cells, ensuring cell size uniformity. This regulation was also observed directly by monitoring nuclear growth in live cells. We also detected cell-size-dependent adjustments of G1 length, which further reduce variability. Combining our assays with chemical/genetic perturbations confirmed that cells employ two strategies, adjusting both cell cycle length and growth rate, to maintain the appropriate size. Additionally, although Rb signaling is not required for these regulatory behaviors, perturbing Cdk4 activity still influences cell size, suggesting that the Cdk4 pathway may play a role in designating the cell's target size.
DOI: https://doi.org/10.7554/eLife.26957.001

*For correspondence:
ran.kafri@sickkids.ca (RK);
marc@hms.harvard.edu (MWK)

Competing interests: The authors declare that no competing interests exist.

## Introduction

Uniformity of cell size is a consistent feature of healthy tissues. While different cell types can differ greatly in size, cells within a given tissue tend to be strikingly similar (*Ginzberg et al., 2015*). In fact, in some tissues, loss of cell size uniformity is a diagnostic marker of malignancy (*Greenough, 1925*). Such observations raise the intriguing question of whether there are dedicated mechanisms that restrict cell size to a specific range. Is size uniformity the product of cellular processes that monitor cell size and correct deviations from a target size (*Ginzberg et al., 2015*; *Lloyd, 2013*)?

Studies of yeast have long postulated the existence of cell-autonomous size control mechanisms. Evidence of cell size checkpoints in the cell cycle of both budding yeast and fission yeast was first reported in 1977 (*Johnston et al., 1977*; *Fantes and Nurse, 1977*). In both species, it was found that the length of the cell cycle is selectively extended in small cells, giving them more time to grow before their next division. Two classes of models have been proposed to explain the biophysical basis of size sensing in yeast (*Wood and Nurse, 2015*). Geometric models propose that cells measure dimensions such as cell length or surface area by relying on highly specific intracellular localizations of cell-cycle regulators such as pom1 or cdr2 (*Martin and Berthelot-Grosjean, 2009*; *Bhatia et al., 2014*; *Pan et al., 2014*). Alternatively, titration-based models suggest that the concentration of a cell cycle activator overcomes the concentration of a cell cycle inhibitor in a growth-dependent manner (*Turner et al., 2012*; *Schmoller et al., 2015*).

Several studies have suggested that the cell cycle of animal cells, like that of yeast, also includes cell size checkpoints (*Killander and Zetterberg, 1965a*; *Dolznig et al., 2004*; *Gao and Raff, 1997*).

**eLife digest** Animal cells come in many different sizes. In humans, for example, egg cells are thousands of times larger than sperm cells. Yet cells of any given type are often strikingly similar in size. The cells that line the surface of organs including the skin and kidneys are especially uniform; in fact a loss of size uniformity in certain tumors is a sign of malignancy. What kind of regulation could enable separate cells within a tissue to have the same size?

One possibility is that each type of cell is programmed with a specific target size, and that a cell can sense if it strays from its target and take steps to compensate. Animal cells sensing their own size was first reported in the 1960s, and now Ginzberg et al. confirm that human cells grown in the laboratory do indeed monitor their size and correct deviations from their target. It turns out that two separate and independent processes help to keep all the cells in the population roughly uniform in size. Firstly, proliferating human cells that are smaller than their target size spend longer growing before they divide. Secondly, at two time points between cell divisions, large cells adjust their growth rate such that they grow slower than small cells.

To show these processes in action, Ginzberg et al. introduced mutations or chemicals that perturbed the length of time between cell divisions or the rate of a cell's growth. As expected, most of these perturbations had only a modest influence on cell size, due to the cell's compensatory strategies. Cells that had less time to grow compensated by more quickly making new protein molecules, meaning that they still had enough material to build two new cells by the time they had to divide. In contrast, if a cell's division was artificially delayed, it reduced its growth rate to stop it from becoming too large. Similarly, cells grown in conditions that slow the production of proteins extended the time between their cell divisions to give them enough time to accumulate the material required for two new cells.

In a recent related study, Liu, Ginzberg et al. identified some of the molecules that a human cell uses to sense its own size. Together these two studies now pave the road to answering a fundamental question in cell biology: what is the elusive cell size sensor? Understanding how cells sense their size will open a window onto how quantitative information is programmed, sensed and communicated within living cells. These findings will shed also new light onto how cells specialize into cell types of different sizes, and what happens when cells lose the ability to sense or regulate their size in diseases like cancers.

DOI: https://doi.org/10.7554/eLife.26957.002

In 1965, Zetterberg and Killander reported that proliferating fibroblasts that have recently entered S-phase are more uniform in size than sister cells that have just exited mitosis (*Killander and Zetterberg, 1965a*). Based on these and other observations, they concluded that cells that are smaller than a critical size are maintained in G1 for longer periods of growth (*Killander and Zetterberg, 1965b*). Subsequent studies have also suggested that size-sensing mechanisms selectively promote S-phase entry in large but not small cells (*Dolznig et al., 2004*; *Gao and Raff, 1997*).

The observation that cell cycle progression is cell-size-dependent raises the question: at which point in the cell cycle does size sensing occur? A defining event in the eukaryotic cell cycle is the '*restriction point*', the time early in G1 when cells commit to undergoing another cell division cycle if growth factors are present (or transition to a quiescent G0 state in the absence of growth factors) (*Pardee, 1974*). In budding yeast, some studies suggest that the restriction point (START) and the cell size checkpoint are one and the same (*Cross, 1995*). In animal cells, reports of size-dependent S-phase entry have suggested that the observed cell size checkpoint may operate later in G1, after the restriction point, leaving this an open question (*Foster et al., 2010*; *Zetterberg and Larsson, 1991*).

Despite the studies described above, research addressing the question of whether animal cells, like yeast, autonomously sense and regulate their own size remains inconclusive (*Lloyd, 2013*). While the work of Zetterberg and Killander focused on the heterogeneous behavior of cells sharing a common extracellular environment, much of the research on animal cell growth focused on extracellular factors that elicit large changes in cell size. These studies, which highlighted pathways controlling cell growth and cell cycle progression that could be differentially activated, led many to conclude

that animal cell growth and cell cycle progression are independent processes (*Conlon et al., 2001*; *Echave et al., 2007*).

Since the 1960s, two alternate models have been proposed to explain cell size homeostasis, the *adder model* and the *sizer model* (*Conlon and Raff, 2003*). According to the adder model, size homeostasis is not the result of size-sensing mechanisms. Instead, size homeostasis is the outcome of a balance between a constant amount of mass that cells accumulate each cell cycle and the reduction in cell mass that accompanies cell division. At the core of the adder model is the assumption that small and large cells accumulate the same amount of mass over the course of the cell cycle. Since large cells lose a greater amount of mass upon division (e.g. half of a large cell is more than half of a small cell), size variation is constrained. In contrast to the adder model, the sizer model assumes that size homeostasis is the product of size-sensing mechanisms that selectively restrict the growth of large cells or promote the growth of small cells.

As the studies mentioned above illustrate, the extent to which the sizer model and adder model describe size homeostasis of animal cells remains unresolved (*Lloyd, 2013*). Furthermore, almost all literature on cell size homeostasis, whether supporting the sizer model or the adder model, is confined to the context of cells that are actively proliferating. In the adder model, size homeostasis is a consequence of cell divisions. In discussions of the sizer model, research has almost exclusively investigated the existence of cell size checkpoints, that is processes that inhibit cell cycle progression for cells that are smaller than a target size (*Wood and Nurse, 2015*). This focus on cycling cells is puzzling since most cells in an animal body are terminally differentiated, do not undergo cell cycles, yet still display size uniformity. In this light, it is compelling to consider the possibility that at least some mechanisms of size homeostasis are not dependent on cell size influencing cell cycle progression, but instead involve an effect of cell size on the rate of cell growth.

One reason questions about cell size control are difficult to answer is that, while it is easy to envision experimental assays of cell size, assays of size *sensing* are more challenging to conceptualize. In this study, we describe new experimental approaches to assay size sensing by monitoring cell size variance. To resolve the ambiguity inherent in our previous approach to this question (*Kafri et al., 2013*), we also develop methods to separately assay the influence of cell size on cell cycle progression and on growth rate. With these approaches, we determine that animal cells monitor their own size and correct deviations in cell size. Our results show that, like yeast, animal cells that are smaller than their target size spend longer periods of growth in G1. Surprisingly, however, we found that in addition to a G1-length extension in small cells, animal cells also employ a conceptually different strategy of size correction. During two distinct points in the cell cycle, anabolic growth rates are transiently adjusted so that small cells grow faster and large cells grow slower. These periods of growth rate adjustment function to lower cell size variability, promoting size uniformity. While cellular growth rates have previously been observed to vary with cell cycle stage (*Kafri et al., 2013*; *Goranov et al., 2009*; *Son et al., 2012*), to our knowledge such a reciprocal coordination of growth rate with cell size has not been previously observed in any organism.

Throughout nature, negative feedback circuits maintain traits and quantities within their appropriate range. Properties such as body temperature and body plan proportions are specified by processes that sense and correct deviations from a target value. The results presented here suggest that the size homeostasis of animal cells is actively maintained by cell-autonomous negative feedback mechanisms that sense and correct aberrations in cell size by adjusting both cell cycle length and growth rate.

## Results

### A size threshold regulates S-phase entry in animal cells

Previous literature has defined cell size as cell volume (*Cadart et al., 2017*; *Tzur et al., 2009*) or cell mass (*Conlon and Raff, 1999*). In this study, we define cell size by a cell's total macromolecular protein mass, as this metric most closely reflects the sum of anabolic processes typically associated with cell growth (*Mitchison, 2003*) and with activity in growth-promoting pathways such as mTORC1 (*Laplante and Sabatini, 2012*). In contrast, cell volume is a more labile phenotype, sensitive to ion channel regulation and fluctuations in extracellular osmolarity.

In *S. cerevisiae*, cell size is thought to be regulated by a cell size checkpoint in G1 (*Johnston et al., 1977*; *Amodeo and Skotheim, 2016*). To test whether a similar size checkpoint functions in animal cells, we made single-cell measurements of cell size, cell cycle stage, and cell age (i.e. time elapsed since last division), in order to investigate the relationship between these properties. We used time-lapse microscopy to image live HeLa and Rpe1 cells for a period of 1–3 days. At the end of the imaging session, cells were immediately fixed and stained with AlexaFluor 647-Succinimidyl Ester (SE-A647), a quantitative protein stain that we previously established as a an accurate measure of cell mass (*Kafri et al., 2013*). This experiment provides three measured properties for each cell. Firstly, movies recorded by time-lapse microscopy reveal the amount of *time* elapsed between each cell's 'birth' via mitosis and its fixation, which we refer to as the cell's '*age*' at the time of fixation. Secondly, staining with SE-A647 reveals each cell's *size* at the time of fixation. SE-A647 staining is sensitive enough to detect the change in cell mass that occurs in less than three hours of growth (less than 15% of the cell cycle) (*Figure 1—figure supplement 1*). Thirdly, using florescent cell cycle indicators, we determine each cell's *cell cycle stage* at the time of fixation.

To quantify '*cell age*' from the recorded time-lapse movies, we customized computational methods to identify cell boundaries, track cell motion, and monitor thousands of individual cells throughout the course of their cell cycle. To determine cell cycle stage, we used cell lines stably expressing mAG-hGem, a fluorescent reporter of APC activation and G1 exit (*Sakaue-Sawano et al., 2008*). Thus, our experimental design independently quantifies age, size, and cell cycle stage for each cell.

This experimental approach allowed us to monitor how the mean cell size changes, as a function of both age and cell cycle stage. A priori, we considered two alternative outcomes, which are depicted in *Figure 1A and B*. If S phase entry is *not* regulated by cell size checkpoints (*Figure 1A*), a cell's size should exclusively reflect the amount of time that it has been growing (i.e. its age), but *not* whether it has passed the G1/S transition. In contrast, if the transition from G1 to S phase is selectively accessible only to cells that have exceeded a critical size threshold (*Figure 1B*), we would expect S-phase cells to be larger on average than cells in G1, even when comparing G1 cells and S-phase cells of identical ages.

The results of this experiment are shown in *Figure 1C and D*. *Figure 1C* shows the average size of HeLa cells as a function cell age. The transient slowing of cell growth about 10 hr after birth (the average age of S-phase entry) is consistent with a behavior we and others have previously identified (*Kafri et al., 2013*; *Gut et al., 2015*). As expected, the average size of the oldest cells is approximately double that of the youngest. Additionally, comparing the size distribution of the youngest cells (first 1.5 hr after birth) to that of mitotics (identified by their rounded shape using phase-contrast microscopy) similarly shows a doubling in cell size over the course of the cell cycle, as expected (*Figure 1—figure supplement 2*). The size distribution of mitotics is identical to that of the oldest cells tracked, confirming that the data in *Figure 1C* represents growth over the course of the entire cell cycle.

In *Figure 1D*, cell size vs. age is plotted separately for G1 and post-G1 subpopulations. Consistent with the existence of a cell size checkpoint (*Figure 1B*), *Figure 1D* shows that S-phase cells are significantly larger than G1 cells of the same age, even though both have been growing for the same amount of time since birth. This suggests that cells exit G1 in a size-dependent manner. Furthermore, the mean size of G1 cells plateaus as cells begins to enter S-phase, consistent with a mechanism where cells enter S-phase upon reaching a particular size threshold. The size difference between identically aged G1 and S-phase cells is statistically significant (student's t-test p<3.92e-04) and was observed in three additional independent replicate experiments (p<0.009, p<5.80e-08, p<9.07e-15).

If cells enter S-phase only after attaining a critical size, cells that are born small are expected to have longer periods of growth in G1. To test this prediction, we assayed the correlation of cell size at birth with a direct measurement of G1-length in single cells. Monitoring the size of live cells is not possible with the SE-A647 staining technique. Instead, we imaged cells with time lapse microscopy and monitored the size of their nuclei (estimated as the area covered by the nucleus in a widefield image), as a proxy for cell size (*Figure 2—figure supplement 1*). In yeast, the nucleus is known to grow continuously throughout all stages of cell cycle and is correlated with cell size (*Jorgensen et al., 2007*). To test whether this is also the case in our experimental system, we measured the correlation between nucleus size and cell size (i.e. SE-A647 intensity) in an unperturbed population of HeLa cells (*Figure 2A*). As expected, this correlation is significant (Pearson's r = 0.68).

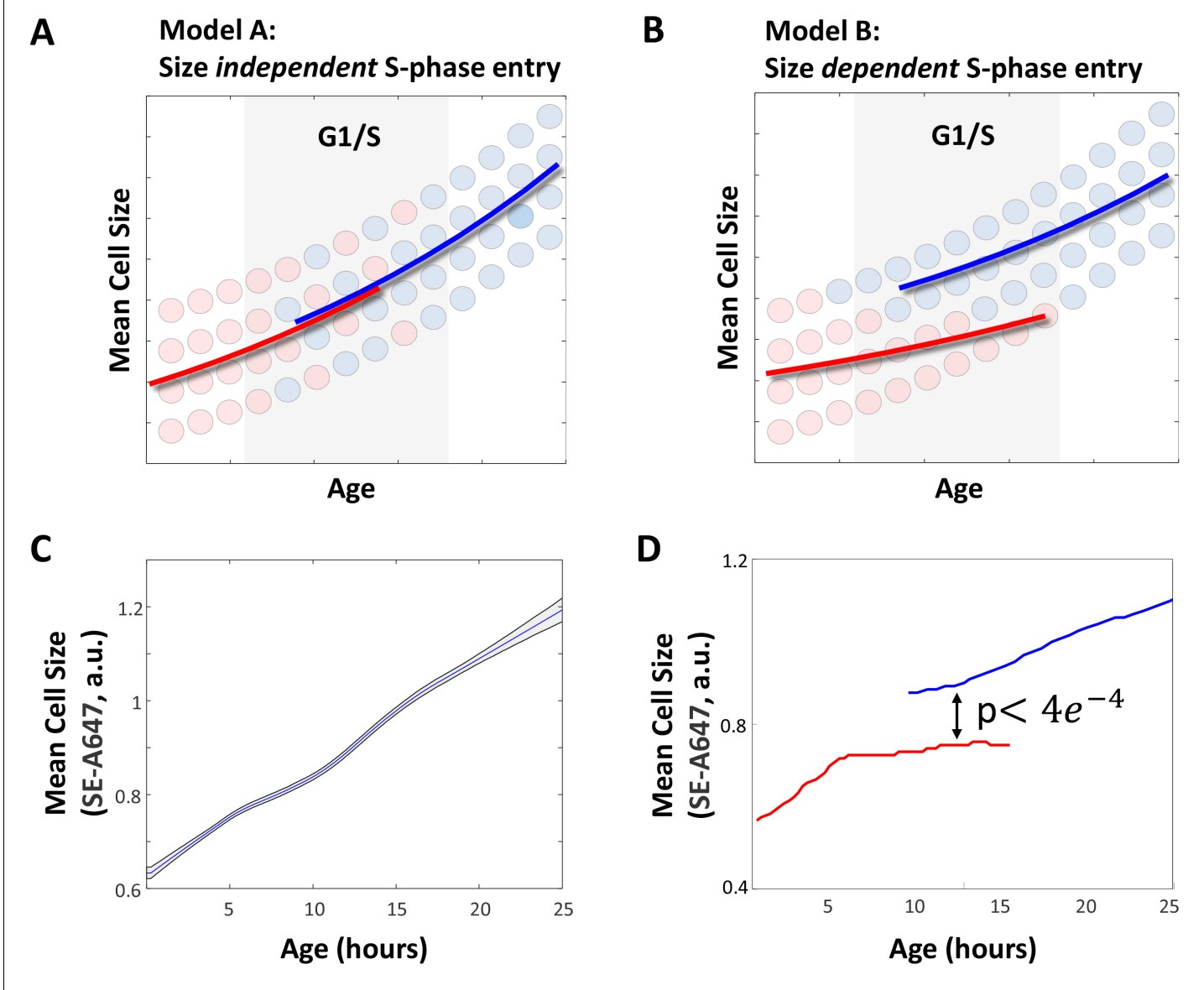

**Figure 1.** Small cells spend more time in G1. (A–B) Two alternate models for the expected behavior of mean cell size as a function of cell age. (A) If G1 exit is independent of cell size, cell size should depend exclusively on cell age, regardless of cell cycle stage. (B) If the G1/S transition is regulated by a cell size checkpoint, cell size should reflect both cell age and cell cycle stage. S-phase cells are expected to have a larger mean size than G1 cells of the same age. We used time-lapse microscopy to image live HeLa and Rpe1 cells for a period of 1–3 days. At the end of the imaging session, cells were immediately fixed and stained with AlexaFluor 647-Succinimidyl Ester (SE-A647). Mean cell size and 'cell age' were determined as described in the Materials and methods as well as the Results sections. (C) Mean cell size as a function of age in HeLa cells. Shaded region marks 90% confidence intervals calculated by bootstrapping. (D) Mean size of G1 (red) and post-G1 (blue) HeLa cells as a function of cell age. Post-G1 cells are larger than G1 cells of the same age (student's t-test p<3.92e-04). The discontinuous leap in average cell size that accompanies the G1/S transition does not imply a discontinuity in the growth curve of individual cells. Instead, differences between the average size of G1 cells and S phase cells are explained by the selective transition of large, but not small cells into S phase (i.e. a cell size checkpoint) as depicted in panel (B) of this figure. Data shown in (C) and (D) are representative examples of four biological replicates (full experimental repeats), n = 5537 cells. The raw data and source code necessary to generate (C–D) are included in *Figure 1—source data 1*.

DOI: https://doi.org/10.7554/eLife.26957.003

The following source data and figure supplements are available for figure 1:

**Source data 1.** File contains the source code (Figure_1.m) and source data necessary to generate *Figure 1C and D* using Matlab.

DOI: https://doi.org/10.7554/eLife.26957.006

**Figure supplement 1.** Protein mass quantification with SE-A647.

DOI: https://doi.org/10.7554/eLife.26957.004

*Figure 1 continued on next page*

*Figure 1 continued*

**Figure supplement 2.** To confirm that the data in *Figure 1C* represent growth over the course of the entire cell cycle, we independently measured the size (SE-A647 intensity) of a sample of mitotic cells identified by their rounded shape and distinctive halo in phase-contrast microscopy.

DOI: https://doi.org/10.7554/eLife.26957.005

*Figure 2B and C* show that both cell size and nucleus size continually increase throughout the cell cycle. *Figure 2C* also highlights that our measurements (both SE-A647 and nucleus size) are sufficiently accurate to resolve the small size increases that accumulate in less than 3 hr of growth (<15% of doubling time). Additionally, the correlation of nuclear size with cell size is strong enough that a plot showing mean nuclear size vs. cell age reproduces the features of the mean cell size vs. cell age curve, including the transient slowing of cell growth around S-phase entry (*Figure 2—figure supplement 2*). Last, we confirmed that growth in nucleus size is independent of DNA synthesis, by treating cells with aphidicolin to prevent DNA replication. Aphidicolin-arrested cells continued to grow in both cell size and nucleus size (*Figure 2—figure supplement 3*), maintaining a correlation between the two.

When we monitored nuclear growth in single cells expressing nuclear-localized cell cycle markers, we observed a significant and reproducible negative correlation between the size of the nucleus at birth and G1 duration (*Figure 2D*). Hela cells that had smaller nuclei at birth spent longer amounts of time in G1. This result, along with the results presented in *Figure 1*, indicates that cells that are smaller at birth compensate with lengthened periods of growth in G1 and is consistent with the model of a cell size threshold at G1 exit. This finding was also independently confirmed in Rpe1 cells, in a separate study in our labs (*Liu et al., 2018*).

While the correlation of nucleus area and G1 length is statistically significant ($p < 6.7 \times 10^{-8}$, *Figure 2D*) and reproducible (N > 4), it is not a strong correlation (Pearson's r = −0.41). One reason for this may be that our measurement of nucleus size is not a perfect correlate of cell size. Also, as we show later in this study, G1 length is not the only means by which cells correct their size. As is elaborated in the discussion, correction of fluctuations in cell size by mechanisms other than cell cycle checkpoints would tend to loosen the demand on the G1-length extension mechanism and weaken the correlation between cell size and G1 length.

To further test whether information about cell size is communicated to the cell cycle machinery, we examined the effect of slowing down cellular growth rates on the length of G1. If cells leave G1 only when a particular size has been reached, slowing down their growth rate would be expected to prolong G1, as cells would require more time to reach the size threshold. Growth rate can be slowed down with drugs such as the mTORC1 inhibitor rapamycin. Culturing Rpe1 cells in 70 nM rapamycin for several days caused a 60% decrease in the average growth rate. As predicted, we observed an 80% increase in the average G1 duration of these cells. Because of the lengthened cell cycle, the profound effect of rapamycin on growth rate caused only a 20% reduction in cell size (*Figure 3A–E*). Previous studies have observed that rapamycin influences both growth rate and G1 length (*Hashemolhosseini et al., 1998*; *Wiederrecht et al., 1995*; *Brown et al., 1994*) and sometimes ascribed those effects to two distinct functions of the mTOR pathway (*Fingar et al., 2004*). Our current results suggest that the influence of rapamycin on G1 length may be an indirect consequence of its influence on growth rate. According to this interpretation, the inhibition of cell growth by rapamycin results in a gradual reduction of cell size which, in turn, triggers compensatory increases in G1 length. This interpretation is consistent with the evidence of cell-size-dependent regulation of G1 length that we observe, even in the absence of rapamycin (*Figure 1D* and *Figure 2D*). Additional support for this interpretation (as well as analogous results in HeLa and other cell lines) will be provided in later sections of this study that examine the timing of rapamycin's effects on cellular growth rate, cell size, and cell cycle progression.

## The rate of cell growth is adjusted in a size-dependent manner

In proliferating cells, cell size is the product of growth duration (cell cycle length) and growth rate, $s = \tau \times v$. Therefore, cells could potentially correct aberrations in size not only by modulating the amount of time ($\tau$) that they grow, by adjusting G1 duration, but also by modulating how fast they grow ($v$). To investigate this possibility, we examined the relationship between cell size and growth rate.

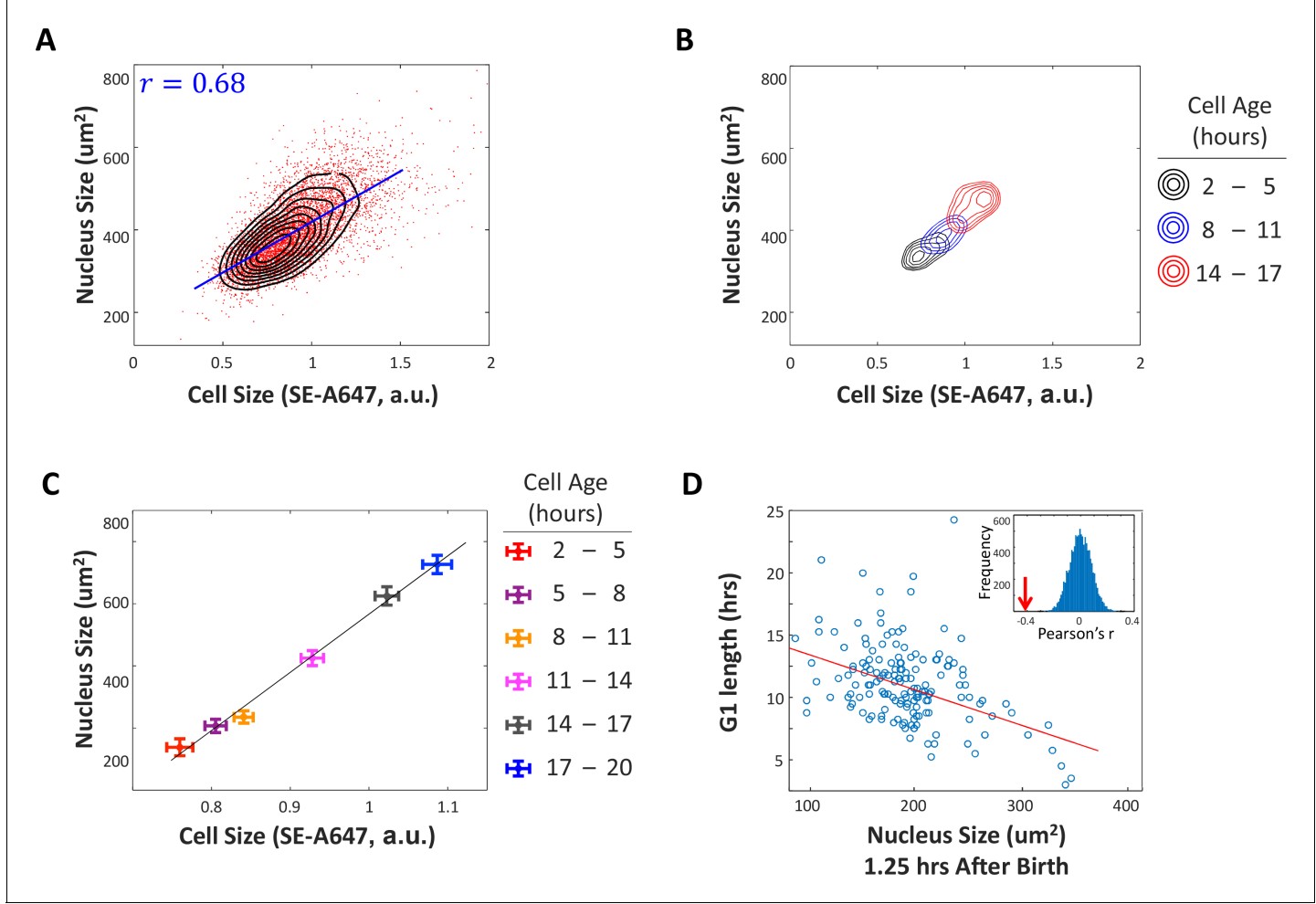

**Figure 2.** G1 length is negatively correlated with nuclear size at birth. (**A**) A significant correlation between cell size and nucleus size suggests that nucleus size is an adequate proxy measurement of cell size. We imaged unsynchronized, unperturbed HeLa cells with time-lapse microscopy and monitored the size of their nuclei (estimated as the area covered by the nucleus in a widefield image), as a proxy for cell size. Datapoints (red) represent single-cell measurements of cell size (SE-A647 fluorescence) and nucleus size (projected area) in a population of HeLa cells. Also shown is the linear regression (blue line) between cell size and the size of the nucleus and the calculated Pearson correlation (r=0.68). Black contour lines represent the calculated joint probability density function, describing the frequency of cells for every given paired value of cell size and nucleus size. (**B**) Joint probability density functions of cell size and nucleus size for cells of different age groups, that is cells 2-5 hours after their most recent cell division (black), cells 8-11 hours after cell division (blue), and cells 14-17 hours after cell division (red). For (**A–C**), single cell measurements of cell age, cell size and nucleus size were performed on a large population of HeLa cells, as described for *Figure 1*. Measurements were performed in multiplex, such that all three parameters were quantified for each individual cell. We then sorted the single cell measurements into bins defined by cell age and separately calculated the joint probability of cell size and nucleus size for each age group. The figure demonstrates that the correlation of cell size and nucleus size consistently persists for cells of different ages. (**C**) Average cell size and nucleus size for cells of different age groups, as described in (**B**). This figure shows that as cells age, they increase in both nucleus size and cell size. Error bars represent standard error calculated as $\frac{std}{\sqrt{n}}$. The significant separation of the datapoints (in relation to errorbars) demonstrates that our cell size measurements and nucleus size measurements are sufficiently accurate to resolve average size differences that result from < 3 hours of growth, approximately 15% of cell cycle duration. (**D**) Size (area covered in widefield fluorescence image) of HeLa nucleus 1.25 hours after birth vs. G1 duration. Line shows least-squares linear fit. Pearson's r = −0.41, p=6.7 x 10$^{-8}$ (p-value calculated using a Student's t distribution for transformation of the correlation). Inset shows distribution of Pearson's r-values generated by randomizing the data, with arrow marking the r-value of non-randomized data. Data shown in (**D**) are representative example of four biological replicates, n=158 cells. The raw data and source code necessary to generate (**A–D**) are included in *Figure 2—source data 1*.

DOI: https://doi.org/10.7554/eLife.26957.007

The following source data and figure supplements are available for figure 2:

**Source data 1.** File contains the source code and source data necessary to generate *Figure 2* and *Figure 2—figure supplement 2* using Matlab.

DOI: https://doi.org/10.7554/eLife.26957.011

**Figure supplement 1.** Monitoring growth of the nucleus in cycling cells.

*Figure 2 continued on next page*

*Figure 2 continued*

DOI: https://doi.org/10.7554/eLife.26957.008

**Figure supplement 2.** Changes in nucleus size mimic changes in cell size.

DOI: https://doi.org/10.7554/eLife.26957.009

**Figure supplement 3.** Nucleus grows with cell during aphidicolin arrest.

DOI: https://doi.org/10.7554/eLife.26957.010

If a cell can sense its own size and adjust its growth rate accordingly – just as a thermostat coordinates heat production with room temperature via short bursts of heat – we might expect these corrections to be transient and subtle. Experimental detection of transient changes in growth rates of single cells has proven to be challenging. To circumvent this difficulty, we derived an indirect inference method to assay whether growth rates of individual cells are dependent on their size. Using the coupled measurements of cell size and cell age described earlier (*Figure 1C*), we calculated the variance in cell size as a function of cell age (i.e. time since division).

In the absence of any size-dependent growth rate regulation, the variance in cell size is expected to increase as cells grow. The reason for this expectation is that individual cells in a population will have some variability in their growth rates, with some cells growing slightly faster than others. The consequence of this cell-to-cell variability is that, in any given time interval, fast-growing cells will accumulate more mass than slow-growing cells, thereby increasing disparities in cell size (*Figure 4A*). In fact, in a synchronized population of growing cells, variance in size can only decrease if small cells grow faster than large cells (*Figure 4B*). To see that this is always true, consider a time interval during which cells grow from $S_1$ to $S_2$. Cell size variance at any given time $t_2$ is related to the cell size variance at any previous time interval, $t_1$, by:

$$Var(S_2) = Var(S_1) + Var(\Delta S) + 2Cov(S_1, \Delta S) \tag{1}$$

where $\Delta S$ is the change in size during the interval, that is the growth rate, and $Cov(S_1, \Delta S)$ is the covariance between initial cell size and growth rate.

Therefore, the change in size variance follows:

$$\begin{aligned} \Delta Var(S) &= Var(S_2) - Var(S_1) \\ &= Var(\Delta S) + 2Cov(S_1, \Delta S) \end{aligned} \tag{2}$$

where $\Delta S$ is the change in cell size accruing over the time period $\Delta t$, that is the growth rate. Since variance is always positive (by definition), cell size variance can decrease with time ($\Delta Var(S) < 0$) if, and only if, the correlation of cell size and growth rate is negative, that is $Cov(S_1, \Delta S) < 0$. This mathematical relationship can be exploited as a method of data analysis. It can be used to detect periods of size-dependent growth rate regulation by analyzing measurements of cell size, without the need to directly measure growth rates.

*Figure 4* shows that periods of decreasing cell size variance do, in fact, occur during the cell cycle of unperturbed HeLa and Rpe1 cells (*Figure 4C–F*). During these periods, cells grow (mean size increases), and yet become more similar in size. This indicates that cellular growth rates are regulated to correct aberrant cell sizes. To quantify the strength of this regulation, we normalized the change in size variance, $Var(S) = Var(S_2) - Var(S_1)$, by the amount of growth that has occurred (i.e. the change in mean size, $\bar{S}$) to define the *coefficient of growth rate variation*, designated $G_{cv}$.

$$G_{cv} = \begin{cases} \dfrac{\sqrt{\Delta Var(S)}}{\bar{S}_2 - \bar{S}_1} & \Delta Var(S) > 0 \\[2ex] -\dfrac{\sqrt{|\Delta Var(S)|}}{\bar{S}_2 - \bar{S}_1} & \Delta Var(S) < 0 \end{cases} \tag{3}$$

The $G_{cv}$-value analysis is a new method to interrogate size control in growing cells. As long as the mean cell size is increasing over time, $G_{cv}$ can be interpreted as follows. If a cell's growth rate is independent of its size, $G_{cv}$ directly equals the coefficient of variation (CV) of cellular growth rates, $G_{cv} = \frac{\sqrt{Var(\Delta S)}}{\bar{S}_2 - \bar{S}_1} = \frac{\sigma_{gr}}{\mu_{gr}}$. This can be shown by substituting the relationship in *Equation 2* into the formula

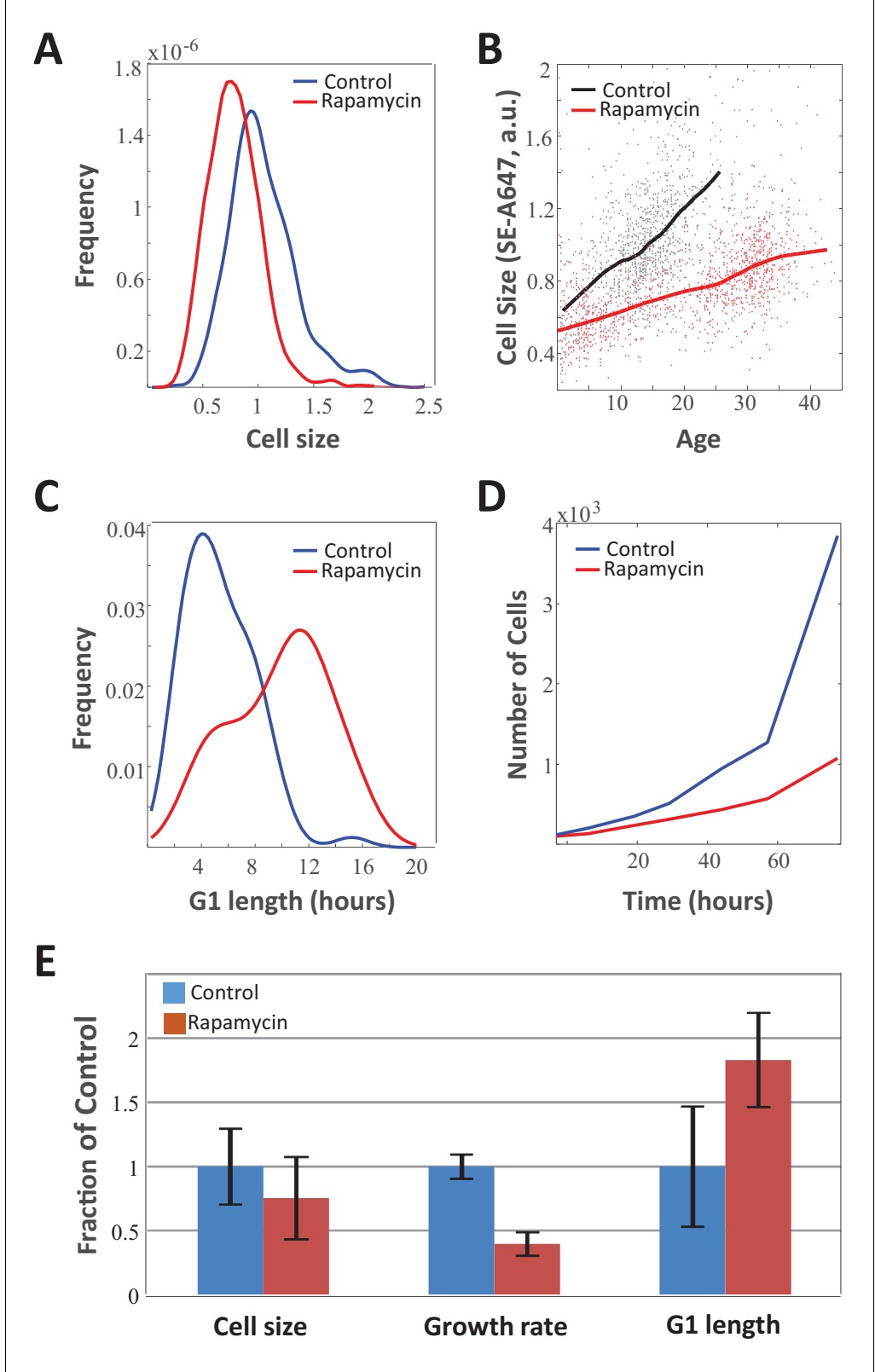

**Figure 3.** Slow growth rate in mTOR-inhibited cells is counteracted by increase in G1 duration. Unsynchronized Rpe1 cells were cultured in the absence or presence 70 nM rapamycin for a period of 3 days. The distribution of (A) cell size (protein content assayed by SE-A647 fluorescence), (B) average growth rate, (C) G1-length, and (D) proliferation rate were obtained as described in the Materials and methods section. Solid lines in (B) show mean

*Figure 3 continued on next page*

*Figure 3 continued*

size vs. age for control and rapamycin-treated cells, while points represent single cell measurements. (**E**) Bar plot comparing the influence of rapamycin on growth rate, G1 length, and cell size. Bar heights represent mean, error bars represent standard deviation (n = 975 cells in control, n = 1268 cells in rapamycin). Data shown are representative examples of two biological replicates (full experimental repeats). The raw data and source code necessary to generate (**A–B**) are included in *Figure 3—source data 1*.

DOI: https://doi.org/10.7554/eLife.26957.012

The following source data is available for figure 3:

**Source data 1.** File contains the source code (Figure_3AB.m) and source data necessary to generate *Figure 3A and B* using Matlab, as well as any necessary functions called by the source code.

DOI: https://doi.org/10.7554/eLife.26957.013

---

for $G_{cv}$ (*Equation 3*), noting that if growth rates are size-independent, $\Delta Var(S) > 0$ and $Cov(S_1, \Delta S) = 0$.

Negative values of $G_{cv}$ (arising from dips in variance) imply that growth rate is actually not independent of size, and that cell size and subsequent growth rate are negatively correlated. We can also note that if $G_{cv}$ is much higher than is plausible for the CV of growth rates (which will equal 1 if growth is a Poisson process), it is likely that growth is positively correlated with size, as in the case of exponential growth.

Plotting $G_{cv}$ versus cell age consistently reveals two distinct periods during which cell size and growth rate are negatively correlated (i.e. $G_{cv}$ is negative), in both HeLa and Rpe1 cells (*Figure 4G, H*). While the exact timing of the drops in $G_{cv}$ varied between experiments (and a third, weaker drop is occasionally observed in HeLa cells), the observation of at least two periods during which $G_{cv}$ is negative is highly reproducible. This result is striking because it suggests communication between cell size and cellular growth rate that is transiently established twice during the cell cycle, perhaps due to cell-cycle-dependent signalling linking size and growth rates.

The analysis presented in *Figure 4* does not discriminate G1 cells from S phase cells and, consequently, the observed drops in variance cannot be explained by the cell-size checkpoint gating G1 (*Figure 4—figure supplement 1*). Therefore, *Figure 4* clearly reveals two distinct times in the cell cycle, where the growth rate of individual cells is selectively repressed in large cells or accelerated in small cells – increasing uniformity in cell size. The only alternative possibility is that cells are removed from the distribution, by either dividing or dying in a size-dependent manner. This is very unlikely to be the case. The dips below zero occur earlier than cells start dividing. (This is true for at least the first dip in Rpe1 and both dips in HeLa, where the mean cell cycle length is 21 hrs, with a standard deviation of 1.6 hrs. A sample cell cycle length distribution is shown in *Figure 4—figure supplement 2*). Furthermore, there was a very low rate of apoptosis in our experiments (<1% of cells imaged from birth died before dividing), so the negative $G_{cv}$ is not explained by cell death.

To test the conclusions of the $G_{cv}$-value analysis, we asked whether the correlation of growth rate and cell size could be observed directly in live cells. Using time-lapse microscopy, we monitored nuclear growth in hundreds of live cells over several days. Comparing growth trajectories collected from the largest and smallest cells in the population provided a dramatic demonstration that growth rates are, indeed, reciprocally coordinated with cell size (*Figure 4I*). Because growth rates became slower in large cells and faster in small cells, individual cells that were initially quite different in size gradually become more uniform in size, explaining the decreases in cell size variance shown in *Figure 4E–H*. Furthermore, as predicted by the $G_{cv}$-value analysis, the negative correlation of cell size and subsequent growth rate was not continuously present, but was transiently established twice during the cell cycle (*Figure 4J* and *Figure 4—figure supplement 3*).

## Growth rate and cell cycle length are coordinated to maintain cell size at fixed values

The $G_{cv}$-value analysis and live-cell tracking all indicate that cellular growth rates are regulated in a manner that reduces cell size variability. Taken together, our results suggest that cells employ two separate strategies to correct deviations from their appropriate size: (1) small cells spend more time in G1 and (2) small cells grow faster than large cells (*Figure 5*). A prediction of this dual-mechanism

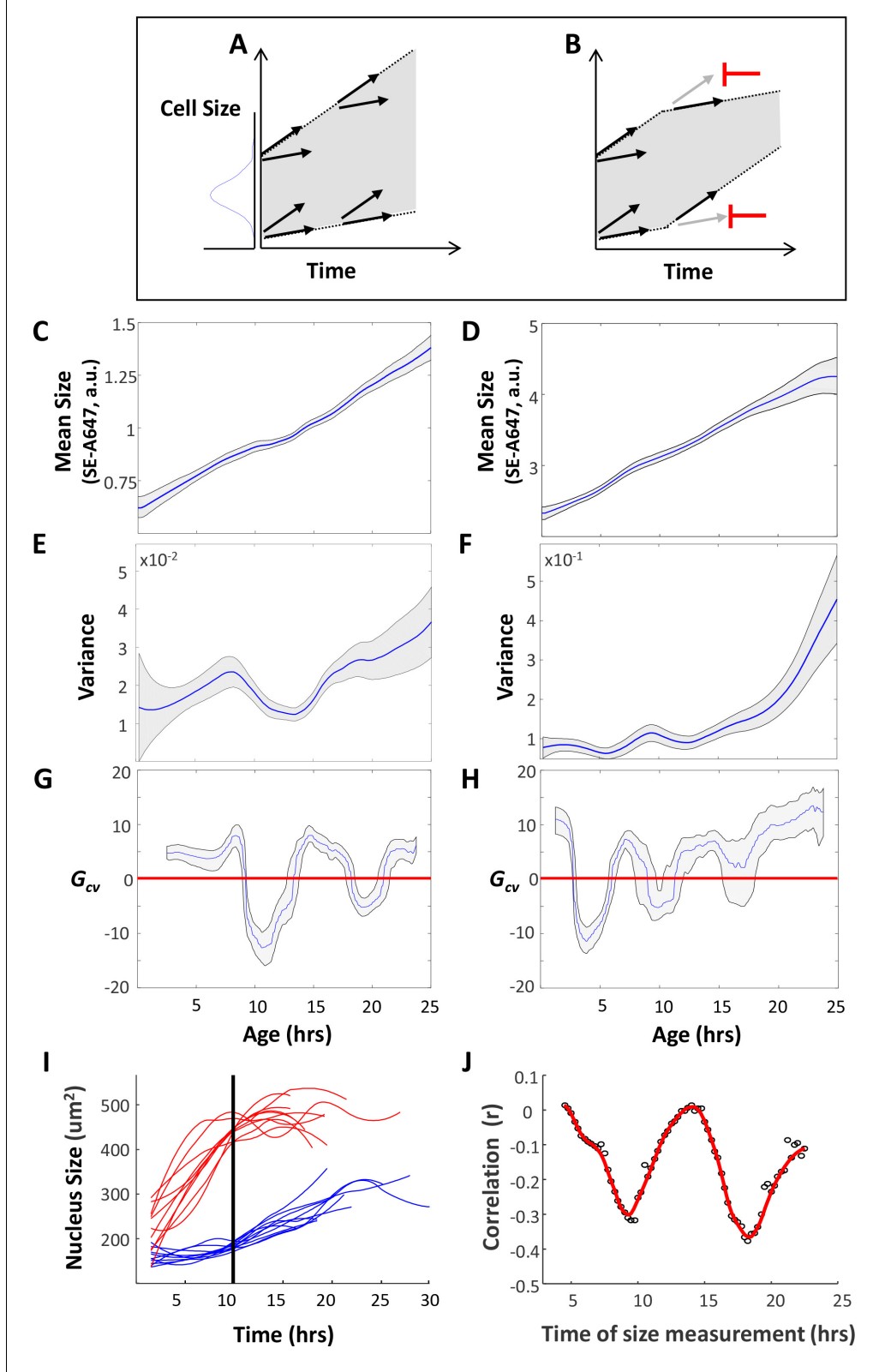

**Figure 4.** Cells adjust their growth rates in a size-dependent manner. (**A**) Variation in growth rates drives an increase in cell size variance over time. (**B**) Regulation that limits cell size variance must function via feedback between cell size and growth in individual cells. HeLa and Rpe1 cells were imaged as described in the legend of *Figure 1*. (**C–D**) Mean cell size (protein content assayed by SE-A647 fluorescence, a.u.) was plotted as a function of age for Rpe1 (**C**) and HeLa (**D**) cells. Shaded region marks 90% confidence intervals calculated by bootstrapping. (**E–F**) Variance in cell size plotted as a function

*Figure 4 continued on next page*

*Figure 4 continued*

of age for Rpe1 (**E**) and HeLa (**F**) cells. Calculation of age-dependent variance was performed as described in Wasserman et al. (***Wasserman, 2010***) as well as in the *Results* section. Shaded region marks 70% confidence intervals calculated by bootstrapping. (**G–H**) $G_{cv}$ plotted as a function of age for Rpe1 (**G**) and HeLa (**H**) cells. Since $G_{cv}$ values are derivatives of the variance plots, minima in the $G_{cv}$ plots are displaced in time, as compared to the minima in the variance plots. Shaded regions mark 50% bootstrapping confidence intervals, and identical trends were seen in all replicates. For Rpe1 plots (**C,E,G**), n = 975 cells, and data shown are representative examples of four biological replicates. For HeLa plots (**D,F,H**), n = 721 cells, and data shown are representative examples of four biological replicates. (**I**) Comparison of growth trajectories of smallest and largest cells. HeLa cells were sorted based on their nucleus size at 10 hr after birth. Nucleus size trajectories for the ten largest (red) and smallest (blue) cells are shown. (**J**) Correlation (Pearson's r) between size and subsequent growth rate (2.5 hr later), calculated as a function of age (time since birth). Data shown in (**I**) and (**J**) are representative examples of four biological replicates, n = 158 cells. The raw data and source code necessary to generate (**C–J**) are included in ***Figure 4—source data 1***.

DOI: https://doi.org/10.7554/eLife.26957.014

The following source data and figure supplements are available for figure 4:

**Source data 1.** File contains the source code and source data necessary to generate ***Figure 4C–J*** using Matlab, as well as any necessary functions called by the source code.

DOI: https://doi.org/10.7554/eLife.26957.018

**Figure supplement 1.** A decrease in cell size variance as a function of cell age reveals cell-size-dependent growth rate regulation.

DOI: https://doi.org/10.7554/eLife.26957.015

**Figure supplement 2.** Cell cycle length distribution of proliferating cells.

DOI: https://doi.org/10.7554/eLife.26957.016

**Figure supplement 3.** Time-lapse imaging confirms that Rpe1 cells adjust their growth rates in a size-dependent manner.

DOI: https://doi.org/10.7554/eLife.26957.017

model is that perturbations that lengthen the cell cycle would be counteracted by a compensatory decrease in growth rate, allowing cells to accumulate the same amount of mass despite the longer periods of growth. Conversely, perturbations that reduce growth rate would be counteracted by a compensatory lengthening of G1.

As an initial test of this prediction, we treated Rpe1 cells with two well-characterized pharmacological inhibitors. To experimentally lengthen the cell cycle, we treated cells with SNS-032, a potent inhibitor of Cdk2. To experimentally decrease growth rates, we used rapamycin, an inhibitor of mTORC1. We optimized working concentrations of both drugs to influence rates of cell cycle progression or growth without causing cell cycle arrest. We then applied the drugs to unsynchronized populations of cells and monitored cell count and average cell size over the course of three days in the presence of the drugs (***Figure 6A***). We fixed samples at various timepoints after the drugs were applied and measured cell size (SE-A647 intensity) and cell count, obtaining a time series of cell size measurements and a corresponding time series of cell count measurements. Cell count was also independently monitored in live samples in each condition, using differential phase-contrast microscopy.

***Figure 6A*** (top panel) shows the influence of the growth rate inhibitor (rapamycin) and the cell cycle inhibitor (SNS-032) on Rpe1 cell size. While SNS-032 treatment caused an *increase* in average cell size, rapamycin treatment caused a *decrease* in average cell size. This result is expected since SNS-032 causes cells to grow for longer periods of time before division and rapamycin causes cells to grow slower. For both drugs, the change in average cell size proceeded

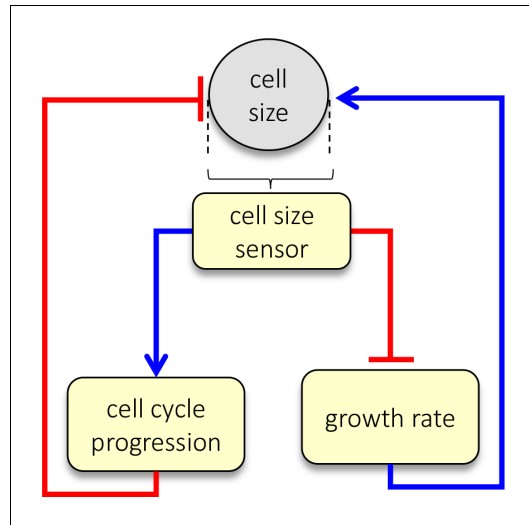

**Figure 5.** Dual-mechanism model of cell size specification. Data presented here are consistent with a model where cells sense their own size and employ two strategies, adjusting both growth rate and cell cycle length, to correct aberrant sizes.

DOI: https://doi.org/10.7554/eLife.26957.019

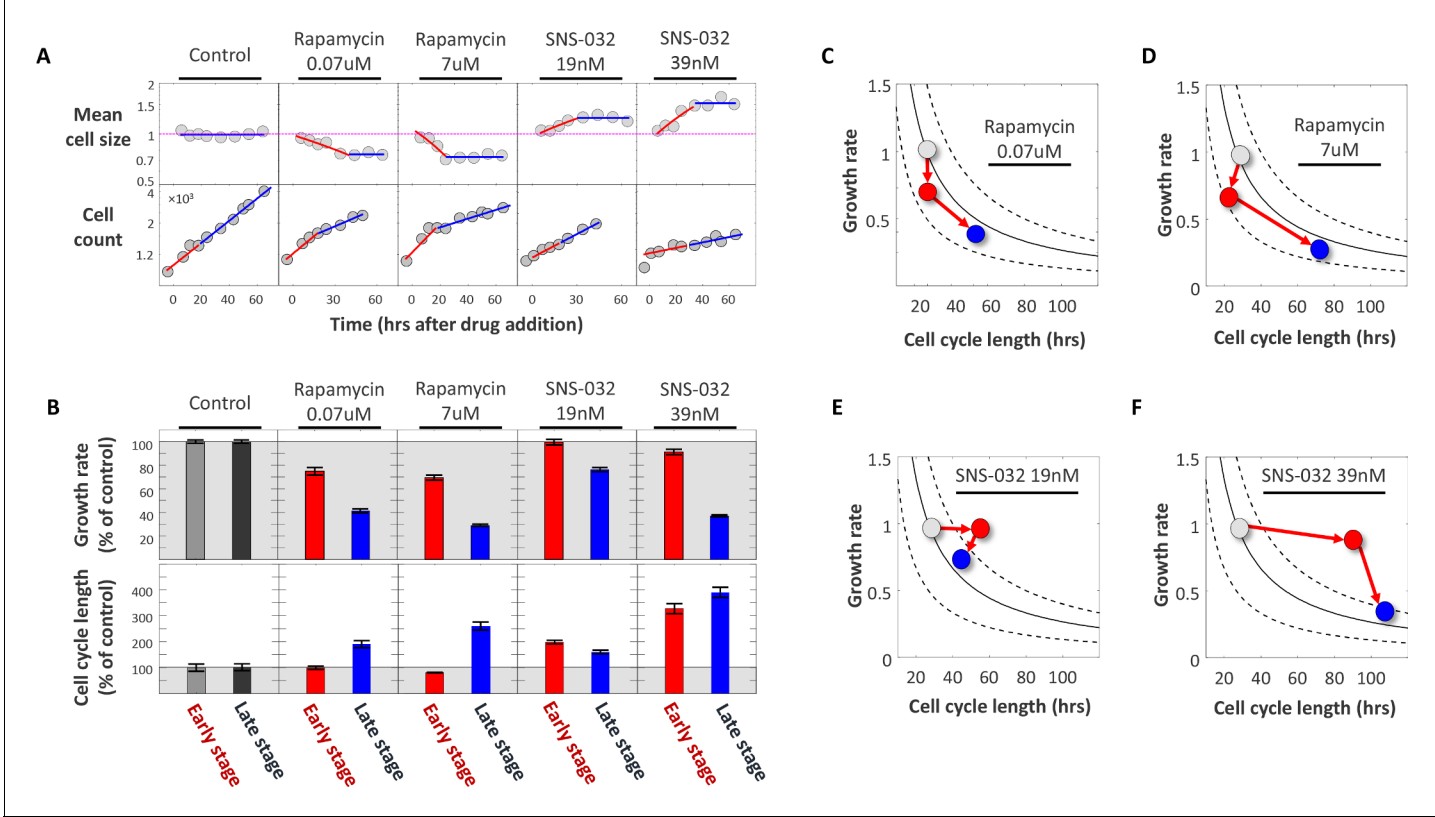

**Figure 6.** Using SNS-032 and rapamycin to examine the coordination of growth rate and cell cycle length. Populations of Rpe1 cells were treated with DMSO (control), rapamycin and SNS-032. Each of the two drugs was used at two different concentrations, as indicated by the plot titles. (**A**) Measurements of average cell size with SE-A647 (top panel) and cell count (bottom panel) were performed (as described in the Materials and methods section) at different time points after drugs were added. Note log scale of y-axes. Each cell size datapoint is an average from a population of >2000 cells. Values of cell size are normalized to control, such that average cell size of the control populations has the value of 1. (**B**) Average growth rate (top panel) and cell cycle length (bottom panel) calculated from data shown in (**A**). Cell cycle length was calculated by fitting exponential curves to measurements of cell count over time shown in the bottom panel of (**A**). For each drug treatment, two separate estimates of growth rate and two separate estimates of cell cycle length were calculated; one for the early stage of the time course (red) and one for the late stage (blue), as described in the text. The separate estimates were calculated by fitting the time course data in (**A**) with two separate regressions, indicated by the blue and red regression curves in (**A**). Values of growth rate and cell cycle length are shown as percent of control. Shaded (gray) area marks growth rate and cell cycle length in the control populations. (**C–F**) Data from (**B**) are plotted as average growth rate versus cell cycle length for each of the separate drug treatments. Data points are color coded with the same color scheme that is used in the bar-plot (**B**), that is colors represent the temporal stages of the drug treatments: gray data-points represent measurements on control populations prior to drug treatment, red data-points represent measurements of populations during the early stage of drug treatment, and blue data-points represent measurements of populations during the late stage of drug treatment. Trend-lines are defined by $v = \frac{C_{TS}}{\tau}$, where $C_{TS}$ is a constant that defines a cell's target size. In other words, all points on a given trend-line represent paired values of growth rate and cell cycle length $(v, \tau)$ that correspond to a single fixed, constant product, $C_{TS}$. An important observation emphasized by panels C-F is that, while drug treatments cause a change in both $v$ and $\tau$, the product, $v \times \tau$, is only temporarily perturbed and returns close to its homeostatic value in the late stage of drug treatment. This is seen by the fact that only the red datapoints fall off the trendlines. Also, the panels clearly highlight the distinction in the order of events that result from cell cycle inhibitors versus growth rate inhibitors, as explained in the text. The raw data and source code necessary to generate (A–F) are included in *Figure 6—source data 1*.

DOI: https://doi.org/10.7554/eLife.26957.020

The following source data is available for figure 6:

**Source data 1.** File contains the source code (Figure_6 .m) and source data necessary to generate *Figure 6* using Matlab.
DOI: https://doi.org/10.7554/eLife.26957.021

only during the initial phase of drug treatment, followed by stabilization of a new cell size that no longer changed with time. Note that the cell size data shown in *Figure 6A* represent population averages. In control populations, while single cells grow and divide, the average cell size remains constant and does not change with time. In drug-treated populations, we observed a gradual

change in average cell size, followed by a plateau (*Figure 6A*). To quantify this trend, we separately considered the early and late stages of the drug treatments. We define the early stage of each treatment as the time interval between drug addition and cell size stabilization (approximately one day), and we define the late stage as the time interval between cell size stabilization and the experiment's end. For the early stage of drug treatment, we used linear regression to characterize cell size as a function of time. For the late stage of drug treatment, average cell size was quantified from the mean (linear regression with a slope of 0). *Figure 6A* (bottom panel) shows the influence of SNS-032 and rapamycin on rates of cell division. As expected, the Cdk2 inhibitor SNS-032 caused an immediate decrease in proliferation rate. Notably, while rapamycin also slowed proliferation rate, this influence only occurred during the late stage of drug treatment. In contrast, rapamycin treatment induced an immediate effect on cell size.

From the measurements of average cell size and cell count over time, two parameters were derived for each sample: average cell cycle length ($\tau$) and average cellular growth rate ($v$). To quantify cell cycle length, $\tau$, measurements of cell count over time were fitted to exponential curves, $N_t = N_0 e^{\alpha t}$ where $N_t$ is cell count at time $t$ and $\alpha = \frac{\ln(2)}{\tau}$. In all cases, fits were constructed with two independent exponential regressions, one estimating the average cell cycle length during the early stage of drug treatment (*Figure 6B*, early time interval) and the second estimating cell cycle length during the late stage of drug treatment (*Figure 6B*, late time interval). This resulted in two estimates of cell cycle length for each drug treatment $(\tau_{early}, \tau_{late})$ as shown in *Figure 6B–F*.

To estimate average growth rate, we relied on the trends of cell count over time and average cell size over time (*Figure 6A*). Specifically, we calculated the rate of increase of the total population's bulk mass and divided that by the cell count. To understand this calculation, consider a simple case where the average cell size does not change over time, as is the case in the control populations that are not treated with drugs. In such populations, even though average cell size does not change with time, bulk biomass ($M_t = cell\ count\ \times\ cell\ size$) does increase, because of the expansion in cell number. To estimate the amount of mass that a single cell accumulates over a short time interval (growth rate), we take the bulk amount of mass ($M_t$) that is accumulated by the *total* population during that interval and divide it by the number of cells in the population. In the general case, we calculate average growth rate with:

$$v = \frac{1}{N_t}\frac{dM_t}{dt} \tag{4}$$

where $M_t$ is given by $M_t = N_t \times \bar{S_t}$ and $N_t$ and $\bar{S_t}$ represent cell count and average cell size at time $t$.

As with the calculation of cell cycle lengths, growth rate values were separately calculated for the early and late stages of drug treatment (*Figure 6B*, *early time interval* and *late time interval*), resulting in two separate growth rate values per drug treatment $(v_{early}, v_{late})$. *Figure 6B–F* show that during the early stage of drug treatment, SNS-032 primarily influences the length of cell cycle, $\tau$, while rapamycin primarily influences the rate of cell growth, $v$. However, later in the drug treatment and after a change in cell size is observed, a coordination of growth rate and cell cycle length becomes apparent. Specifically, while SNS-032 caused an immediate lengthening of cell cycle, its influence on growth rate was observed only after prolonged drug treatment (*Figure 6B,E and F*). This suggests that the influence of cdk2 inhibitor SNS-032 on growth rate is indirect and is mediated by a property that accumulates over time, presumable cell size. Conversely, rapamycin caused an immediate reduction in growth rate, while its influence on cell cycle length occurred much later (*Figure 6B–D*). This delayed effect of rapamycin on cell cycle length is directly apparent from the abrupt change in the slope of the proliferation curves of rapamycin treated cells (*Figure 6A*, bottom panel, rapamycin treatments).

The sharp contrast between the temporal order of events that follow rapamycin and SNS-032 treatment is illustrated in *Figure 6C–F*, where growth rate is plotted as a function of cell cycle length for each time interval. The curves overlaid on *Figure 6C–F* delineate paired values of growth rate and cell cycle length that correspond to a fixed product, $\tau \times v = constant$. Data points that fall close to a given curve represent conditions that, while different in growth rate and cell cycle length, will yield cells of similar size. Comparing the measurements shown in *Figure 6C–F* to these iso-size curves shows that the compensatory adjustments of cell cycle length or growth rate seen during the late stage of rapamycin and SNS-032 treatments, respectively, counteract the initial effects of the

drugs and return the product $\tau \times v$ to its homeostatic value. This explains the relatively small influence of these drugs on cell size (depending on drug dose, size decreases by 20–30% in rapamycin, and increases by 25–50% in SNS-032), compared to their effects on growth rate and cell cycle length.

## The inverse coordination of growth rate and cell cycle length is robust to a variety of chemical perturbations

A quantitative prediction of the dual-mechanism model shown in *Figure 5* is that growth rate, $v$, and cell cycle length, $\tau$, are coordinated to maintain cell size at its fixed target size, $\tau \times v = \mathit{target\ size}$. This means that growth rate and cell cycle duration are related by:

$$v_i = \frac{C_{TS}}{\tau_i} \tag{5}$$

where $v_i$ and $\tau_i$ represent the growth rate and cell cycle length in experimental condition $i$, while $C_{TS}$ is a constant that defines the cell's target size. To test the generality of this prediction, we repeated the experiment described in *Figure 6* using a variety of cdk inhibitors to slow cell cycle progression, as well as several inhibitors of cell growth. Each drug was used at several different concentrations where cells maintain viability and proliferation. We used a simplified version of the procedure described above for rapamycin and SNS-032 (without separating the time series into early and late stages) to calculate the average growth rate and cell cycle length of Rpe1cells during a three-day incubation in each drug.

*Figure 7A and B* show that most chemical perturbations of growth rate and cell cycle length had surprisingly small influences on Rpe1 cell size. Whiles drugs produced up to 4-fold changes in both growth rate and cell cycle length, the largest changes in cell size were close to 1-fold (i.e. 50% change in cell size), with most treatments changing size by less than 30%. Consistent with the dual-mechanism model, treatment with all but one (palbociclib, *Figure 8A*) of the tested compounds resulted in coordinated changes of both cell cycle length and growth rate, such that cell size remained relatively unchanged. Also, consistent with the results in *Figure 6*, growth rate inhibitors caused a small decrease in cell size (*Figure 7A*, rightmost panel) while cell cycle inhibitors caused a small increase in cell size (*Figure 7B*, rightmost panel). Lastly, separate analysis of early and late stages of the drug treatments confirmed that inhibitors of cell cycle regulators cause immediate lengthening of the cell cycle followed by compensatory decreases in growth rate, while cycloheximide, torin-2, and rapamycin induce immediate decreases in growth rate followed by compensatory increases in cell cycle length (*Figure 7—figure supplement 1*).

Of the tested cdk inhibitors, only the cdk4/6 inhibitor palbociclib increased cell cycle length without triggering a compensatory decrease in growth rate, causing an unusual increase in cell size (*Figure 8A,B*). Knockdown of cdk4 and cdk6 by siRNA yielded a similar increase in cell size (*Figure 8—figure supplement 1*). This result shows that, while the coordination of growth rate and cell cycle length is robust, it is dependent on mechanisms that can be perturbed or reprogrammed. In an attempt to further disrupt the processes that maintain cell size uniformity, we treated Rpe1 cells with the HSP90 inhibitor radicicol. HSP90 is known to suppress phenotypic variability (*Hsieh et al., 2013*) and has also been found to regulate both cell cycle progression (*Mollapour et al., 2010*; *Bandura et al., 2013*) and mTORC1 mediated growth in response to cellular stress (*Conn and Qian, 2011*). Remarkably, culturing cells in radicicol not only decoupled growth rate and cell cycle length (*Figure 8A,B*), but also increased cell size variability by 26% (±3.2%) relative to untreated cells (see Materials and methods). In contrast, cdk4/6 inhibition with palbociclib had no significant influence on cell size heterogeneity. This finding further illustrates that cell size uniformity is a regulated phenotype that can be perturbed by disrupting the coordination of growth and cell cycle progression.

*Figure 9A* shows that the reciprocal influence of the various drug treatments on growth rate and cell cycle length in Rpe1 cells match the trend predicted by *Equation 5*. Changes in growth rate, $v$, and cell length, $\tau$, are not only reciprocally related, but also quantitatively coordinated such that the product, $v \times \tau$, is relatively constant. To test the generality of the compensation mechanism, these measurements were repeated, in four additional cell lines: U2OS (*Figure 9B*), SAOS2 (*Figure 9C*), HeLa (*Figure 9D*) and 16HBE (*Figure 9E*). Because cell size is sensitive to perturbation of Cdk4 (*Figure 8A,B*), we deliberately included cell lines that lack Rb signaling (Rb-null SAOS2, as well as

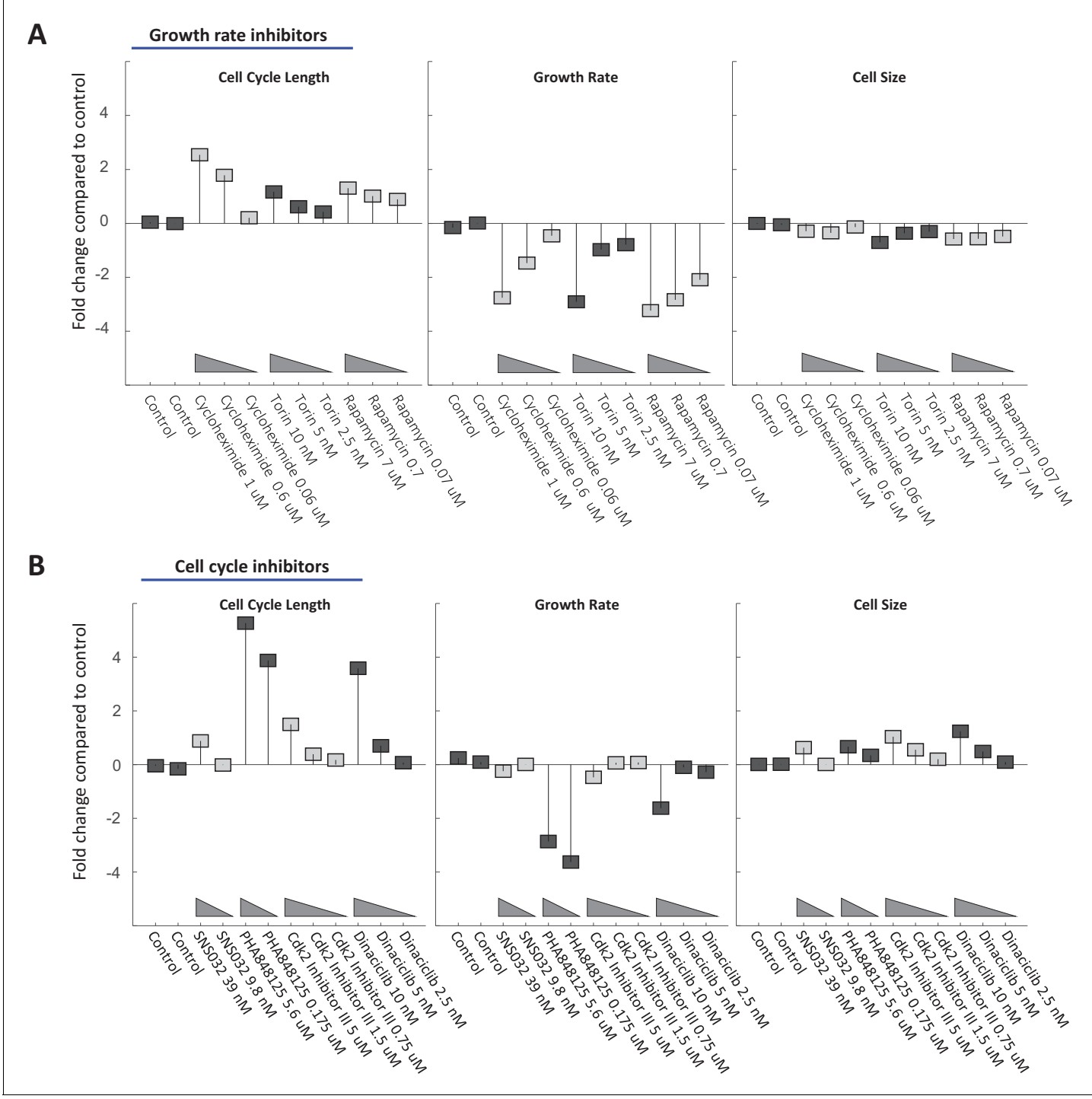

**Figure 7.** Cell size is stabilized by a compensatory relationship between cell cycle length and growth rate. Rpe1 cells were treated with varying doses of drugs inhibiting growth (**A**) or cell cycle progression (**B**). The fold change in mean cell cycle length (left panel), growth rate (middle panel), and cell size (right panel), relative to untreated cells, is plotted for each condition. The mean cell cycle length and mean growth rate of cells in each condition were calculated from measured increases in bulk protein and number of cells over the course of a 68 hr incubation, as described in the text. Mean cell size (protein content assayed by SE-A647 staining) was measured after 44 hr of incubation in each condition. To better characterize their influence, all drugs were applied at a range of concentrations, as indicated by the titles in the x-axes. The drug concentrations shown were found to be non-toxic and meet the following two criteria in preliminary tests: (i) They cause a detectable effect on growth rate and/or cell cycle progression. (ii) They do not block proliferation entirely. Each drug treatment was done in duplicate, alongside six control (DMSO) samples, with several thousand cells in each sample. Fold-changes are shown relative to the average cell cycle length, growth rate, and cell size of all six control samples. The raw data and source code

*Figure 7 continued on next page*

*Figure 7 continued*

necessary to generate (**A–B**) are included in *Figure 7—source data 1*. The drugs used and their targets are as follows: mTOR inhibitors (Torin and Rapamycin), protein synthesis inhibitor (Cycloheximide), cdk1/2 inhibitors (SNS-032, cdk2 Inhibitor II, PHA848125, Dinaciclib).

DOI: https://doi.org/10.7554/eLife.26957.022

The following source data and figure supplement are available for figure 7:

**Source data 1.** File contains the source code (Figure_7 .m) and source data necessary to generate *Figure 7* using Matlab.

DOI: https://doi.org/10.7554/eLife.26957.024

**Figure supplement 1.** Drug-induced changes in cell cycle length or growth rate are followed by compensatory changes.

DOI: https://doi.org/10.7554/eLife.26957.023

HeLa and 16HBE which are expected to have disrupted Rb activity [*Landry et al., 2013*; *Ahuja et al., 2005*]) alongside cell lines with intact Rb signaling (Rpe1 and U2OS). Since the cell lines are differentially sensitive to the various drugs, drug concentrations were optimized separately for each cell line to find doses that cause a detectable change in cell cycle length and/or growth rate but don't cause cell death or cell cycle arrest. The conditions shown in *Figure 7* are represented by the gray circles in *Figure 9A*. For *Figure 9B–E*, the drug concentrations used, along with their effects on growth rate, cell cycle length, and cell size are shown in *Figure 9—figure supplement 1– 4*.

As expected, palbociclib caused large increases in cell size in Rpe1 and U2OS cells, but not in the Rb-inactive cell lines. (Note that, since drug doses were optimized to cause an effect, the palbociclib doses shown in *Figure 9C and E* are high (2–4 μM, *Figure 9—figure supplements 1,2*), so the response of 16HBE cells may be due to an off-target effect. The Rb-inactive cell lines were completely insensitive to the palbociclib doses (60–500 nM) used on Rpe1 and U2OS cells.) Aside from the differences in palbociclib sensitivity, however, *Figure 9A–E* show very similar trends. The intact size compensation in Rb-inactive cell lines suggests that, while the cyclin D-Rb axis plays a role in specifying the target cell size, this pathway is not responsible for the coordination of growth rate with cell cycle length to maintain cell size uniformity.

## Shortening cell cycle length by overexpression of cyclin E, but not cyclin D, leads to an increased growth rate that maintains size homeostasis

An interesting finding derived from *Figures 7–9* is the difference in the way cell size is affected by perturbations of Cdk1/2 and Cdk4/6. Inhibitors of Cdk1/2 and inhibitors of Cdk4/6 both cause an increase in the duration of cell cycle. However, this lengthening of cell cycle is compensated by decreased growth rates in the case of Cdk1/2 inhibitors, but not in the case of the Cdk4/6 inhibitor, palbociclib (*Figures 7–9*). In classical descriptions, Cdk2 and Cdk4 partner with cyclin E and cyclin D, respectively, to promote the G1/S transition. To explore whether cyclin E and cyclin D play divergent roles in the regulation of cell size, we generated two stable cell lines that overexpress either cyclin E1 or cyclin D1 upon induction with doxycycline. As a positive control, we also generated a cell line with doxycycline-inducible overexpression of p27, which is expected to lengthen (rather than shorten) the cell cycle. Inducible expression was validated by western blotting (*Figure 10—figure supplement 1*). As expected, overexpression of both cyclin D and cyclin E resulted in higher proliferation rates (i.e. shorter cell cycles), while overexpression of p27 resulted in slower proliferation rates (longer cell cycles) as compared to control (*Figure 10A,B,C*). Curiously, despite the similar influence of cyclin E and cyclin D overexpression on cell cycle lengths (*Figure 10A,B*), these genetic perturbations had markedly different influences on cell size (*Figure 10D,E*). While overexpression of cyclin E shortened the cell cycle, the size distribution of cells overexpressing cyclin E was indistinguishable from control cells (*Figure 10A,D*), due to a compensatory *increase* in growth rate (*Figure 9A*). Similarly, while overexpression of p27 lengthened the duration of cell cycle, cells overexpressing p27 were only minimally different in size (*Figure 10C,F*). In contrast, overexpression of cyclin D caused a shortening of cell cycle that was accompanied by dramatic reduction in cell size (*Figure 10B,E*). The ability of cells to compensate for genetic perturbations of cyclin E but not for perturbations of cyclin D is consistent with our pharmacological perturbations and suggests a distinct role for cyclin D/Cdk4

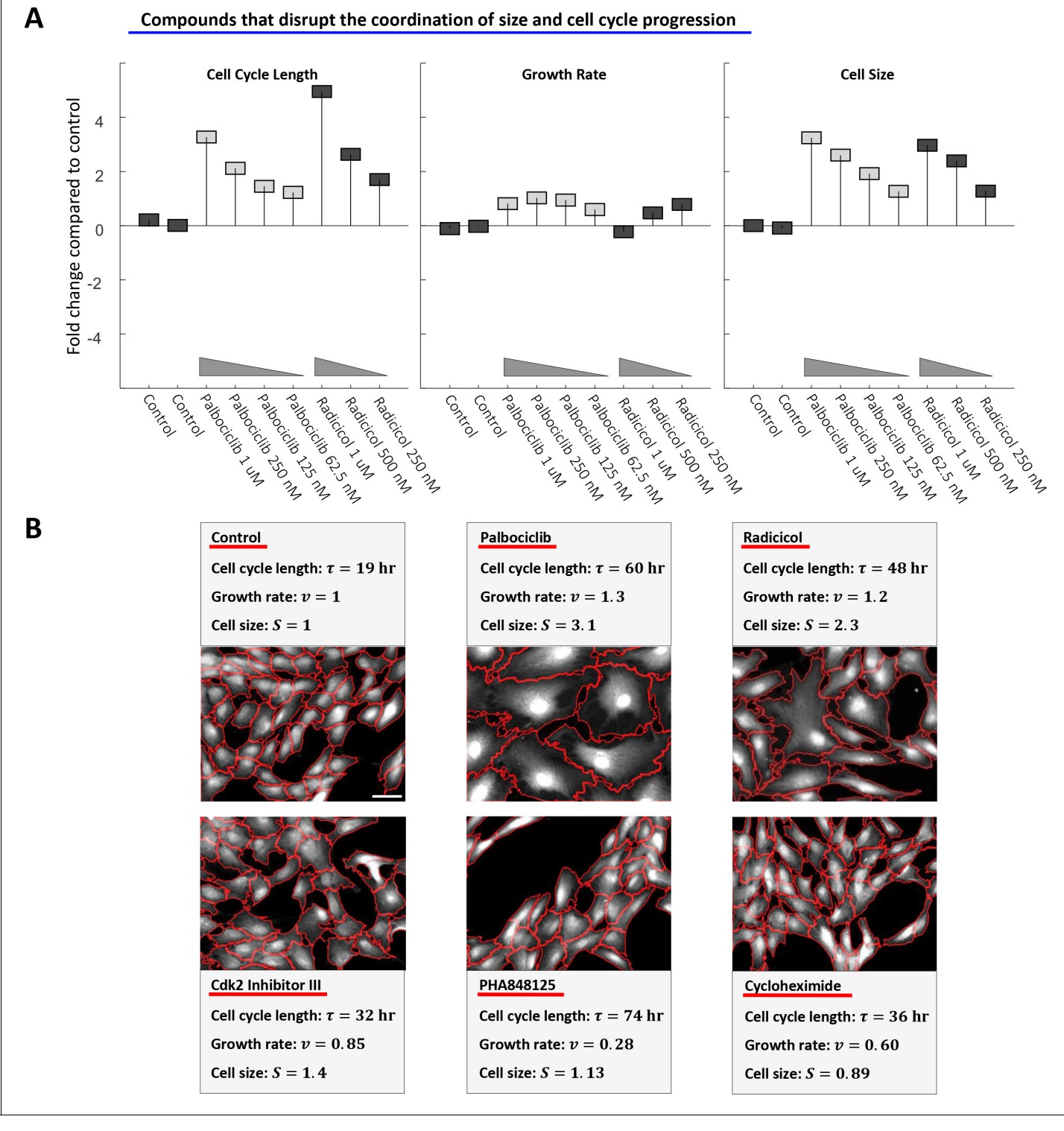

**Figure 8.** Perturbing the relationship between cell cycle length and growth rate. (A) Rpe1 cells were treated with varying doses of palbociclib (cdk4/6 inhibitor) or radicicol (hsp90 inhibitor). (A) The fold change in mean cell cycle length (left panel), growth rate (middle panel), and cell size (right panel), relative to untreated cells, is plotted for each condition. The mean cell cycle length and mean growth rate of cells in each condition were calculated from measured increases in bulk protein and number of cells over the course of a 68 hr incubation, as described in the Materials and methods and Results sections. Mean cell size (protein content assayed by SE-A647 staining) was measured after 68 hr of incubation in each condition. Each drug treatment was done in duplicate, alongside six control (DMSO) samples, with several thousand cells in each sample. Fold-changes are shown relative to the average cell cycle length, growth rate, and cell size of all six control samples. (B) Representative widefield fluorescence images (SE-A647 stain) of

*Figure 8 continued on next page*

*Figure 8 continued*

unperturbed cells and cells treated with 600 nM cycloheximide, 175 nM PHA848125, 5 uM Cdk2 Inhibitor III, 500 nM palbociclib, or 500 nM radicicol. All images are shown at the same scale, scale bar = 50 um. Red lines delineate cell boundaries. The raw data and source code necessary to generate (A) are included in *Figure 8—source data 1*.

DOI: https://doi.org/10.7554/eLife.26957.025

The following source data and figure supplement are available for figure 8:

**Source data 1.** File contains the source code (Figure_8 .m) and source data necessary to generate *Figure 8A* using Matlab.

DOI: https://doi.org/10.7554/eLife.26957.027

**Figure supplement 1.** siRNA knockdown of CDK4 and CDK6 causes an increase in cell size.

DOI: https://doi.org/10.7554/eLife.26957.026

in target size regulation. (Note that this result differs from that obtained in drosophila, where cyclin D/Cdk4 overexpression did not affect proliferating cell size (*Datar et al., 2000*).)

## Discussion

To date, the subject of cell size uniformity has remained largely unexplored. In this study, we use multiple experimental approaches (*Figures 1–4*) to show that cells that have escaped their appropriate size range are corrected by two separate mechanisms: regulation of G1 length and regulation of cellular growth rate (*Figure 5*). To test this model, we show that chemical or genetic perturbations that alter cell cycle length consistently lead to reciprocal changes in growth rate, such that cell size remains constant (*Figure 6 Figure 7B*). Similarly, perturbations that change growth rate lead to reciprocal changes in cell cycle length (*Figure 6* and *Figure 7A*). In both cases, the compensatory changes occur after prolonged drug treatment (*Figure 6* and *Figure 7—figure supplement 1*), once the initial effects of the drugs have yielded a change in cell size, supporting the model illustrated in *Figure 5*. To further test whether cell size is buffered from changes in cell cycle length, we overexpressed cyclin E or p27 to shorten or lengthen the duration of the cell cycle, respectively. In both cases, we found that the changing the cell cycle length resulted in a compensatory increase (cyclin E) or decrease (p27) in growth rate to keep cell size constant (*Figure 10*).

To directly visualize the negative correlation of cell size with G1 length, we used time lapse microscopy to show that cells that are born with a smaller nucleus spend longer periods of growth in G1 (*Figure 2D*). While the correlation of cell cycle length and nucleus size is statistically significant and reproducible, there remains the question of why this correlation is small in magnitude. One possibility is suggested by the model in *Figure 5*. Since deviations from target size are corrected not only by G1-length extension but also by adjustments to the rate of cell growth, the burden on each one of those separate mechanisms is weakened. In support of this idea, we repeatedly observe that when growth is inhibited with rapamycin the slope of nucleus size versus G1 length becomes about 50% steeper (See Liu et. al., *Figure 3G* versus 3I [*Liu et al., 2018*]).

While the possibility that the size of animal cells is regulated by cell cycle checkpoints has been previously debated, the possibility that feedback selectively suppresses growth rate in large cells is new and has only been sparsely explored (*Kafri et al., 2013*). Accurately measuring the correlation of growth rate and cell size is challenging, due to the difficulty of measuring the growth rate of individual cells with high sensitivity. To circumvent this challenge, we developed the $G_{cv}$-value analysis to infer this correlation from measurements of cell size variance. With this approach, we not only provide the first definitive proof that individual animal cells autonomously monitor their own size and adjust their growth rate accordingly, but also provide new experimental assays for the study of growth rate regulation.

The possibility that size homeostasis is maintained by growth rate regulation may have broad implications. In animals, most cells are terminally differentiated. These cells do not divide and yet, often undergo precise adjustments of cell size (*Nie et al., 2013*). Also, it is often in such terminally differentiated tissues that size uniformity is most striking. This suggests the presence of size specification mechanisms that are not dependent on regular cell divisions. Furthermore, the size threshold model is fundamentally limited in its ability to correct the size of very large cells. If some cells grow fast enough to double even in a very short cell cycle, regulation of cell cycle length alone cannot constrain size variability in the population.

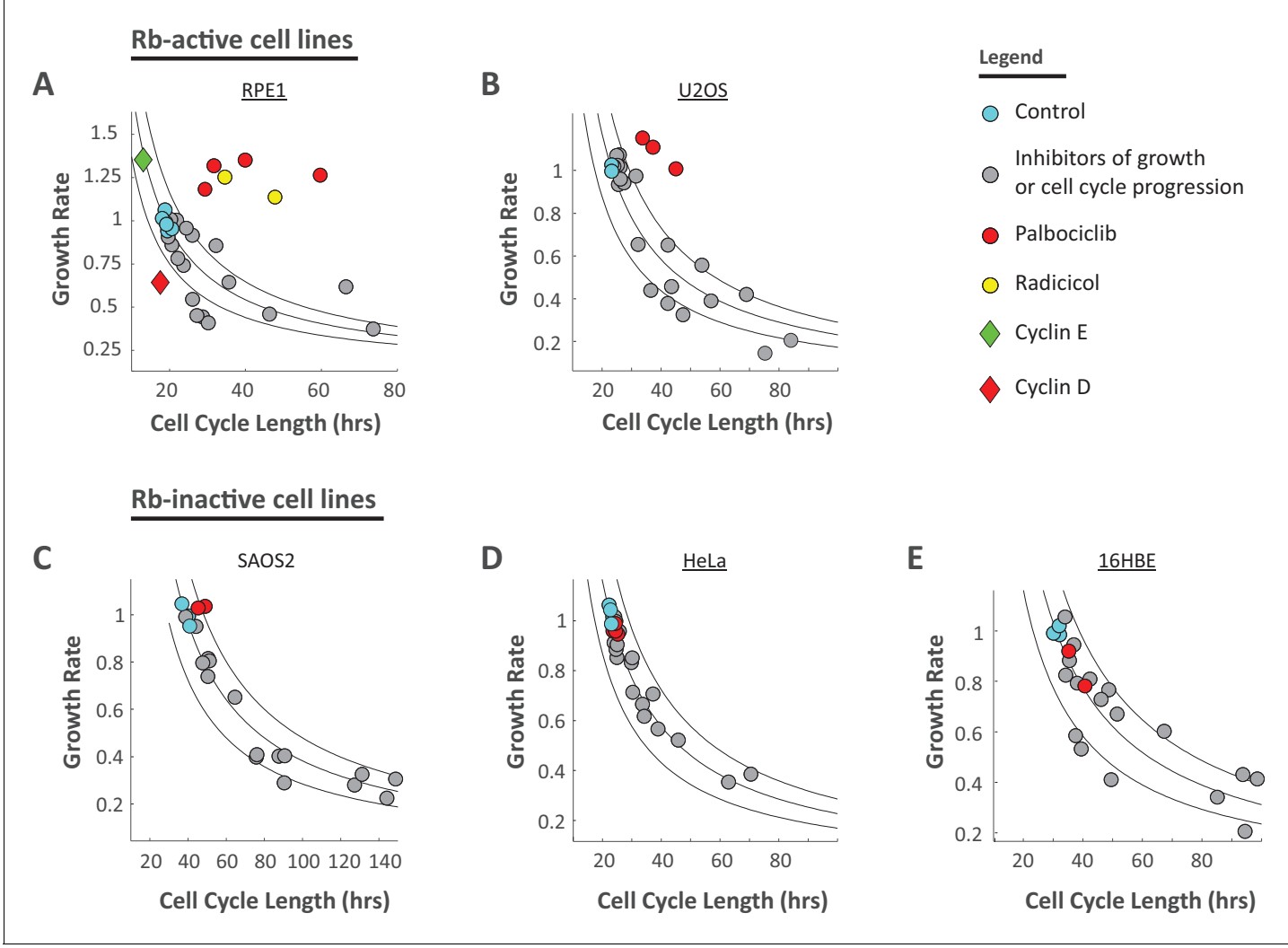

**Figure 9.** Compensatory relationship between cell cycle length and growth rate. (**A**) Rpe1 cells were treated with varying doses of drugs inhibiting growth or cell cycle progression, as described in the legends of **Figures 7** and **8**. Mean growth rate vs. mean cell cycle length are plotted for cells in each condition shown in **Figure 7**. Cyan circles represent unperturbed cells (DMSO). Gray circles represent the inhibitors of growth or cell cycle progression shown in **Figure 7A and B**. Red circles (palbociclib) and yellow circles (radicicol) represent the conditions that disrupt the coordination of cell growth and cell cycle progression shown in **Figure 8A**. Mean growth rate vs. mean cell cycle length are also plotted for stable Rpe1 cell lines inducibly overexpressing cell cycle regulators (diamonds). Green diamond represents overexpression of cyclin E1 (average of four replicates, several thousand cells per sample), and red diamond represents overexpression of cyclin D1 (average of three replicates, several thousand cells per sample). Black lines mark iso-cell-size contours. Region between lines spans a 25% shift in cell size. Calculation of the Pearson correlation coefficient between log (growth rate) and log(cell cycle length), using data from conditions marked by gray circles (inhibition of growth or cell cycle progression) and green diamond (cyclin E1 overexpression), yields r = −0.76, p=2.1×10$^{-7}$, by two-tailed Student's t-test, indicating a significant inverse correlation between growth rate and cell cycle length in these conditions. (**B–E**) The experiment presented in **Figure 7** and **Figure 8A** was repeated in several additional cell lines. The resulting mean growth rate vs. mean cell cycle length plots are shown for U2OS (**B**), SAOS2 (**C**), HeLa (**D**), and 16HBE (**E**). The corresponding plots of fold change in cell cycle length, growth rate, and size in each condition, for these four cell lines, are shown in **Figure 9—figure supplements 1–4**. For (**A–E**), growth rates are measured in arbitrary units of accumulated SE-A647 intensity per hour. In each experiment, all calculated growth rates were divided by a scaling factor such that the average growth rate of control samples = 1, to aid in comparison between cell lines. The raw data and source code necessary to generate (**A–E**) are included in **Figure 9—source data 1**.

DOI: https://doi.org/10.7554/eLife.26957.028

The following source data and figure supplements are available for figure 9:

**Source data 1.** File contains the source code and source data necessary to generate **Figure 9** and its associated figure supplements, using Matlab.
DOI: https://doi.org/10.7554/eLife.26957.033

**Figure supplement 1.** U2OS cell size is stabilized by a compensatory relationship between cell cycle length and growth rate.
DOI: https://doi.org/10.7554/eLife.26957.029

*Figure 9 continued on next page*

*Figure 9 continued*

**Figure supplement 2.** SAOS2 cell size is stabilized by a compensatory relationship between cell cycle length and growth rate.
DOI: https://doi.org/10.7554/eLife.26957.030

**Figure supplement 3.** HeLa cell size is stabilized by a compensatory relationship between cell cycle length and growth rate.
DOI: https://doi.org/10.7554/eLife.26957.031

**Figure supplement 4.** 16HBE cell size is stabilized by a compensatory relationship between cell cycle length and growth rate.
DOI: https://doi.org/10.7554/eLife.26957.032

A conclusion of this study is that the *size* of an animal cell is maintained at its *target size value* by homeostatic coordination of growth rate and growth duration. These results propose a conceptual

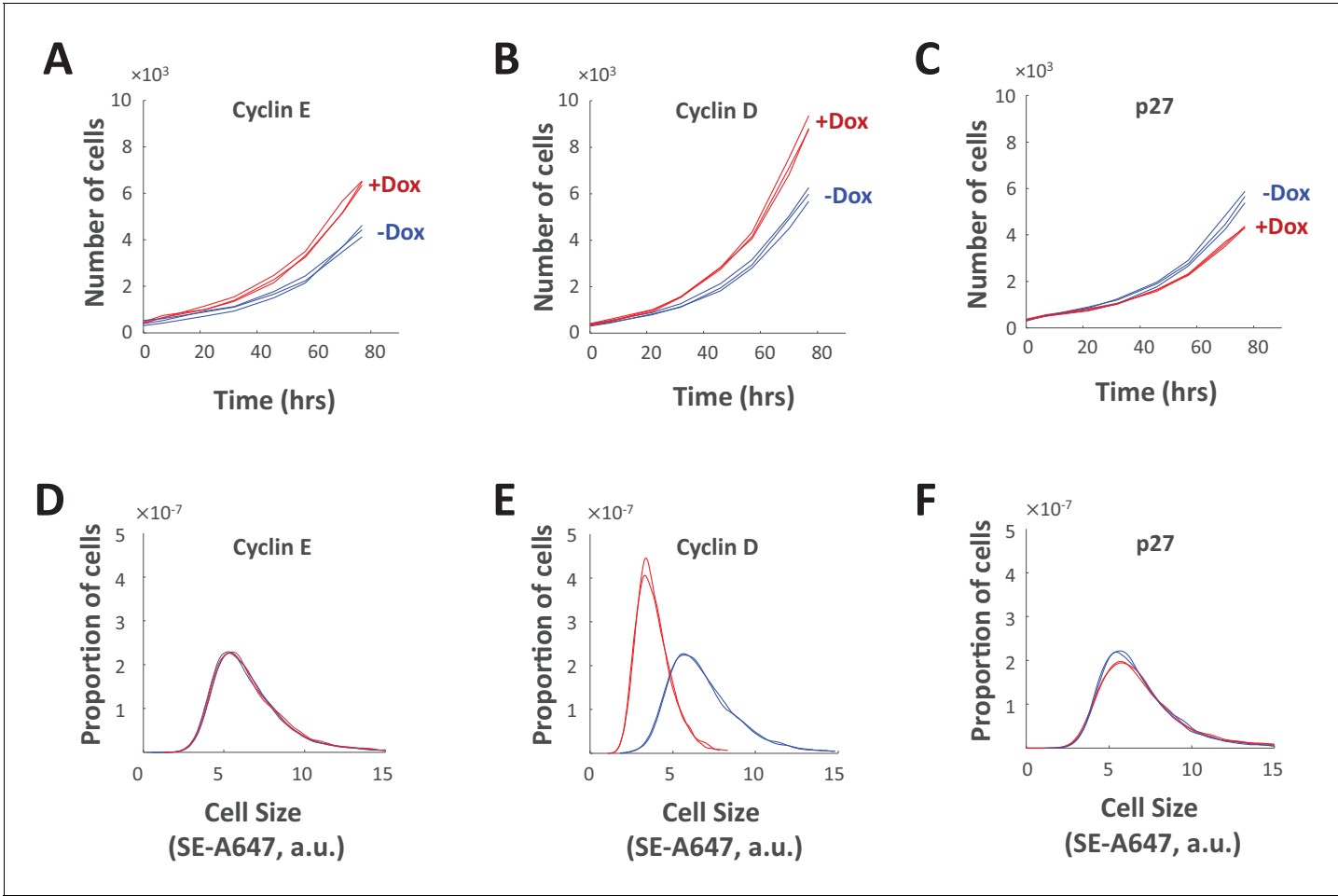

**Figure 10.** Genetic perturbations of cell cycle length. (A–C) Proliferation of Rpe1cells with doxycycline-induced overexpression of cyclin E1 (A), cyclin D1 (B), or p27 (C). For (A–C), proliferation was monitored by differential phase contrast imaging of untreated cells (blue) and doxycycline-treated cells (red). (D–F) Cell size distributions of Rpe1 cells after 77 hr of doxycycline-induced overexpression of cyclin E1 (D), cyclin D1 (E), or p27 (F). For (D–F), cell size was measured by SE-A647 staining of untreated cells (blue) and doxycycline-treated cells (red). The experiment was done in triplicate, and cell size was measured in several thousand cells per sample. The raw data and source code necessary to generate (A–F) are included in *Figure 10—source data 1*.
DOI: https://doi.org/10.7554/eLife.26957.034

The following source data and figure supplement are available for figure 10:

**Source data 1.** File contains the source code (Figure_10 .m) and source data necessary to generate *Figure 10* using Matlab.
DOI: https://doi.org/10.7554/eLife.26957.036

**Figure supplement 1.** Doxycycline-inducible overexpression of cyclin E, p27, and cyclin D1.
DOI: https://doi.org/10.7554/eLife.26957.035

distinction between a cell's *actual size* and a cell's *target size,* in the sense that a cell can temporarily be larger or smaller than its target size. Cells that are smaller than their target size grow faster and for longer periods of time. This finding raises the question of what mechanism dictates the cell's target size. While this study, and Liu et al., (*Liu et al., 2018*) show that cell size homeostasis is maintained by changes in both the rate and the duration of cell growth (*Figures 6*, *7*, *9* and *10*), we also show that cell size is sensitive to perturbations of Cdk4/cyclin D (*Figures 8* and *10*). Both pharmacological inhibition of Cdk4 and genetic overexpression of cyclin D cause significant changes in cell size value. These results are also consistent with the dramatic influence of Cln3 (a cyclin D homologue) on cell size in budding yeast (*Tyers et al., 1992*). Altogether, these results suggest that cell size is strongly influenced by the Rb/cyclin D axis. In *Figure 9*, we show the coordination of growth rate with cell cycle length in five different cell lines, including Rb-inactive cell lines as well as cell lines with intact Rb signaling. To our surprise, measurements of Rb positive and Rb negative cell lines produced similar results. This contrast between the strong influence of cyclin D/Cdk4 on cell size to the dispensability of Rb for size homeostasis may be telling. On one hand, *Figure 9* shows that Rb is not required for maintenance of size homeostasis, nor for the coordination of growth rate with cell size length. On the other hand, the dramatic influence of cyclin D/Cdk4 on cell size (*Figure 8*, *Figure 10*) implies an important role for the Rb-cyclin D axis in size regulation. An intriguing possibility is that the function of the Rb-cyclin D axis is in dictating the specific cell size value (i.e. 'target size') to be maintained rather than in ensuring cell size homeostasis at that target size. Consistent with the possibility that Cdk4 is involved in target size specification, palbociclib, a cdk4/6 inhibitor, causes a relatively uniform increase in cell size, suggesting a reprograming of target size. In contrast, the size increase caused by radicicol, an hsp90 inhibitor, is accompanied by an increase in the variability of cell size, suggesting a disruption of cell size homeostasis. While this evidence is not sufficient to determine whether cyclin D/Cdk4/Rb is involved in target size specification, it does warrant further explanation in a future study.

The results presented here refute the premise that cell size is simply the quotient of independently regulated rates of cell growth and division (*Lloyd, 2013*). Instead, we show evidence of feedback between these two processes that robustly maintains cell size even in the face of strong perturbations of either growth or cell cycle progression. The presence of feedback that maintains a constant cell size, and minimizes variation in cell size, suggests the presence of a cell size sensor. With an understanding that this sensor influences growth and cell cycle progression in opposite ways, and with assays that can separately quantify each mode of regulation, we are in a position to uncover its molecular identity.

## Materials and methods

### Cell culture

Cells were grown in Dulbecco's Modification of Eagles Medium (Cellgro, 10–017-CV), supplemented with 10% Fetal Bovine Serum (FBS, Gibco, 26140) and 5% Penicillin/Streptomycin (Cellgro 30–002 CI), incubated at 37°C with 5% CO2. Measurements were made when cells were 50–75% confluent, to avoid the effects of sparse or dense culture on cell growth and proliferation.

### Cell cycle markers

To distinguish S-phase cells from cells in G1, we used cell lines stably expressing two nuclear-localized fluorescent cell cycle markers: (1) the geminin degron fused to Azami Green (mAG-hGem) and (2) the cdt1 degron fused to monomeric Kusabira Orange 2 (mKO2-hCdt1) (*Sakaue-Sawano et al., 2008*).

### Time-lapse microscopy

Cells were seeded in 6-well, glass-bottom (No. 1.5) plates one day prior to imaging. Cells were seeded at low densities such that they would not reach confluence during the course of the experiment, as described in Materials and methods-Cell culture. Leibovitz's L-15 Medium (ThermoFisher, 21083027) with 10% FBS was used during image acquisition, with a layer of mineral oil on top of the media to prevent evaporation. The microscope was fitted with an incubation chamber warmed to 37°C. Widefield fluorescence and phase contrast images were collected at 15 min intervals on a

Nikon Ti motorized inverted microscope, with a Nikon Plan Fluor 10 × 0.3 NA objective lens and the Perfect Focus System for maintenance of focus over time. A Lumencor SOLA light engine was used for fluorescence illumination, and a Prior LED light source was used for transmitted light. A Prior Proscan III motorized stage was used to collect images at multiple positions in the plate during each interval. Images were acquired with a Hamamatsu ORCA-ER cooled-CCD camera controlled by MetaMorph software. After two days of time-lapse imaging, cells were fixed and stained as described below. Images of fixed cells were acquired on the same microscope, at the same stage positions, and an image-registration tool was designed in Matlab to match individual cells in the time-lapse movies to the corresponding cells in the fixed images.

## Fixation and staining
Cells were fixed in 4% paraformaldehyde (Alfa Aesar, 30525-89-4) for 10 min, then permeabilized in chilled methanol for 5 min. Cells were stained with 0.04 ug/mL Alexa Fluor 647 carboxylic acid, succinimidyl ester (SE-A647, Invitrogen A-20006), to label protein. DNA was stained with DAPI (Sigma D8417).

## Pharmacological perturbations
To slow growth rate the following drug treatments were used: cycloheximide (1, 0.6, and 0.06 uM, Sigma C4859), Torin-2 (10, 5, and 2.5 nM, Tocris 4248), rapamycin (7, 0.7, and 0.07 uM, CalBiochem 553211). To slow the cell cycle, the following drug treatments were used: BN82002 (25, 12.5, 6.2, 3.1, 1.6, and 0.78 uM, Calbiochem 217691), SNS-032 (39 and 9.8 nM, Selleckchem S1145), PHA848125 (175 nM, Selleckchem S2751), Cdk2 Inhibitor III (5, 1.5, 0.75, and 0.38 uM, Calbiochem 238803), Dinaciclib (10, 5, and 2.5 nM, Selleckchem S2768), Palbociclib (500, 250, 125, and 62.5 nM, Selleckchem S1116).

Radicicol (Tocris, 1589) treatments were done at concentrations of 250 nM, 500 nM, and 1 uM.

Each drug treatment was done in duplicate, alongside six control (DMSO-treated) samples, with several thousand cells in each sample.

Cells were imaged using a Perkin Elmer Operetta high content microscope, controlled by Harmony software, with an incubated chamber kept at 37°C and 5% $CO_2$ during live-cell imaging. A Xenon lamp was used for fluorescence illumination, and a 740 nm LED light source was used for transmitted light. To monitor proliferation of live cells, differential phase contrast images were collected every 6–12 hr, over the course of three days, using a 10 × 0.4 NA objective lens. A plot of the number of cells vs. time was fitted to an exponential growth model to calculate the average cell cycle time.

Samples were fixed after 14, 44, and 68 hr in each drug. After fixation and staining with SE-A647 (protein) and DAPI (DNA), fluorescence images were collected with a 20 × 0.75 NA objective lens. The bulk protein content (total SE-A647 intensity of sample) and number of cells were measured in each sample. From these measurements, we calculated the average growth rate and cell cycle length of cells in each condition, by fitting all data points (from two replicates of each condition) to an exponential growth model. Cell cycle length was also independently measured by monitoring the proliferation of live cells in each condition with differential phase-contrast microscopy, as described above.

## Genetic perturbations – Overexpression of cell cycle regulators
Entry clone vectors were obtained encoding open reading frames for CCND1 (clone V57299 from Openfreezer), CCNE1 (clone HsCD00045400 from DNA Resource Core, Harvard Medical School) and CDKN1B (clone HsCD00080627 from DNA Resource Core, Harvard Medical School). To alter cell cycle length by genetic manipulation, we generated stable Rpe1 cell lines with doxycycline-inducible overexpression of cyclin D1, cyclin E1, or p27 (CDKN1B) using the Retro-X Tet-One Inducible Expression System (Clontech 634307) as per manufacturer's instructions. At the start of each experiment cells were treated with doxycycline at concentrations of 1000 ng/mL (for cyclin D1 and p27 overexpression), 100 ng/mL (for cyclin E overexpression shown in *Figure 8*), or 50 ng/mL (for cyclin E overexpression shown in *Figure 9*). Doxycycline concentrations were optimized for maximum effect on proliferation in each cell line. Proliferation was monitored by differential phase contrast imaging over the course of a 77 hr incubation in doxycycline, with periodic fixation and staining

of samples to measure average growth rate and cell cycle length in each condition, as was done for the pharmacological perturbations described above. Each doxycycline induction was done in triplicate, alongside three untreated control samples of the same cell line. For comparison with drug-treated Rpe1 cells (*Figure 8A*), growth rates and cell cycle lengths of each cell line were normalized so that the untreated controls of each cell line matched the average non-transduced Rpe1 control sample, since the engineered cell lines had slightly different basal proliferation rates.

## Genetic perturbations – siRNA knockdown of cell cycle regulators

ON-TARGETplus SMARTpool siRNAs for the genes of interest (Dharmacon L-003238–00, L-003240–00, L-003236–00) as well non-targeting negative control siRNAs were obtained from Dharmacon (Lafayette, CO). DharmaFECT 1 Transfection Reagent (Dharmacon T-2001) was used to transfect Rpe1 cells with each siRNA or combination of siRNAs, as indicated in *Figure 8—figure supplement 1*. Cells were harvested at 48 and 72 hr post-transfection, for western blot verification of siRNA knockdown. Cells were also fixed and stained with SE-A647 at these time points, to measure cell size as described above.

## Image analysis and data analysis

All image analysis (cell segmentation, tracking, measurements of fluorescence intensity and nuclear size) and data analysis was performed with custom-written tools in Matlab. The mean and variance of cell size as a function of age (*Figure 1C* and *Figure 4C–H*) were calculated by nonparametric regression, as described by Wasserman (*Wasserman, 2010*). To quantify the influence of drugs (palbociclib and radicicol) on the variation in cell size, we used the normalized median absolute deviation (MAD), that is $\frac{MAD(x)}{median(x)}$.

## Monitoring nuclear growth in live cells

We quantified the 2-d projected area of the nucleus (i.e. the area of the image covered by the nucleus), which we found correlates well with cellular protein content (*Figure 2A–C* and *Figure 2—figure supplement 2*). As illustrated in *Figure 2—figure supplement 1*, we tracked individual HeLa cells over time and monitored their cell cycle progression and nuclear growth. Note that, in these cells, the nucleus tends to elongate as it grows, rather than expanding as a sphere. (This is significant because spherical expansion could yield a spurious correlation between size and growth of the 2-d projected area). We chose to monitor projected area so as not to make assumptions about nuclear shape in calculating volume, while bearing in mind the potential artifacts. We also trimmed the first 1.25 hrs of each trajectory, to avoid the effects of possible changes in nuclear shape as the cell flattens after mitosis. Trajectories ended at the onset of mitosis, when the nuclear envelope breaks down.

## Acknowledgements

Microscopy was done at the Nikon Imaging Center at Harvard Medical School, with the help of Jennifer Waters and Lara Petrak. Celina Qi assisted in computational analysis of time-lapse movies. Justin Sing assisted in western blot validation of engineered cell lines. We would like to thank Yuval Dor for helpful discussions and advice. Work was supported by the Canadian Institutes of Health Research (grant FRN-343437 to RK) and the National Institute of General Medical Sciences (grant GM26875 to MK), as well a National Science Foundation Graduate Research Fellowship (MBG). We also thank Patricia and Alexander Younger and the Younger foundation for their generous donation to support our research.

## Additional information

### Funding

| Funder | Grant reference number | Author |
| --- | --- | --- |
| National Science Foundation | Graduate Research Fellowship | Miriam Bracha Ginzberg |

| Canadian Institutes of Health Research | FRN-343437 | Ran Kafri |
| National Institute of General Medical Sciences | R01GM026875 | Marc W Kirschner |

The funders had no role in study design, data collection and interpretation, or the decision to submit the work for publication.

### Author contributions

Miriam Bracha Ginzberg, Conceptualization, Data curation, Software, Formal analysis, Validation, Investigation, Visualization, Methodology, Writing—original draft, Project administration, Writing—review and editing; Nancy Chang, Investigation, Helped generate the cell lines used for the experiments shown in Figure 10 and Figure 9A; Heather D'Souza, Investigation, Helped with the measurements of cell size, growth rate, and cell cycle length in response to perturbations (these measurements contributed to the generation of Figure 6, Figure 9B,C,E, and Figure 10); Nish Patel, Investigation, Methodology, Writing—review and editing; Ran Kafri, Conceptualization, Resources, Software, Formal analysis, Supervision, Funding acquisition, Investigation, Visualization, Methodology, Writing—original draft, Project administration, Writing—review and editing; Marc W Kirschner, Conceptualization, Resources, Supervision, Funding acquisition, Project administration, Writing—review and editing

### Author ORCIDs

Miriam Bracha Ginzberg ⓘ http://orcid.org/0000-0001-8431-3714
Ran Kafri ⓘ http://orcid.org/0000-0002-9656-0189
Marc W Kirschner ⓘ https://orcid.org/0000-0001-6540-6130

### Decision letter and Author response

Decision letter https://doi.org/10.7554/eLife.26957.039
Author response https://doi.org/10.7554/eLife.26957.040

## Additional files

### Supplementary files

• Transparent reporting form
DOI: https://doi.org/10.7554/eLife.26957.037

### Data availability

All data presented in this study are included in the manuscript and supporting files. Source data files have been provided for all figures.

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
