## [Decision Letter]

Thank you for submitting your article "Cell size sensing in animal cells coordinates growth rates and cell cycle progression to maintain cell size uniformity" for consideration by *eLife*. Your article has been favorably evaluated by Marianne Bronner (Senior Editor) and two reviewers, one of whom, Bruce Edgar (Reviewer #1), is a member of our Board of Reviewing Editors.

The reviewers have discussed the reviews with one another and the Reviewing Editor has drafted this decision to help you prepare a revised submission.

Summary:

In this manuscript, the authors report a fundamental new insight into the nature of cell size homeostasis in human cells, namely that cells have an active mechanism to reduce size variation at specific points in the cell cycle. Evidence for this mechanism is obtained by the real time image analysis of thousands of cells over ~24-72h, which allows prior growth history, G1 exit and/or nuclear volume to be correlated with bulk protein content as a proxy for cell mass. Analysis of the variation in growth in both HeLa and RPE1 cells demonstrates that size variation decreases at two specific times after cell birth, a result that can be explained by an active size compensation mechanism whereby large cells grow more slowly and small cells grow more rapidly. In addition, the authors also provide further evidence for the previously suspected mechanism of a delay in G1 phase progression to compensate for reduced growth, arguing for two distinct mechanisms of active size control. The response of the homeostatic system to perturbation of either the cell cycle or growth machinery is probed using a series of different chemical inhibitors, with the surprising conclusion that the system is robust to all inhibitors, except for a CDK4/6 inhibitor and an HSP90 inhibitor. Overall, the discovery of a compensation mechanism that actively reduces size variance is an important new finding in the size control field. Although the manuscript does not address the molecular mechanism(s) for this compensation mechanism, the discovery itself stands on its own merit without the need for mechanistic insight at this stage. There is no doubt that further work will be spurred on by this study, which poses previously unimagined questions about size control.

Essential revisions (from the reviews):

1) The Introduction needs revision.

(from review #1): The Introduction fails to outline the "sizer" models proposed, and experimentally supported for budding and fission yeast. These should be explained, as they are the best examples of size control in the literature. In addition I would disagree with the authors statement that the field believes that "growth and cell cycle progression are not coordinated." On the contrary, there's massive evidence showing that cell growth regulates cell cycle progression. Please re-write the intro to better reflect the literature and beliefs in the field.

(from review #2): The authors could do a better job framing the size control problem, perhaps starting by describing the observations of Zetterberg et al. briefly. I don't agree with the claim that the consensus view is that size control doesn't exist in animal cells. The weight of evidence for size control going all the way back to Zetterberg is strong, and the review cited to the contrary is an outlier opinion that isn't by any means representative, and thus I don't think it is appropriate to quote from this review. The fixation on the general issue of active size control to some extent distracts from the key finding that there is a bona fide size-dependent compensation mechanism, as opposed to a simple threshold. On this note, the authors make no mention all of the restriction point, which would seem a logical candidate for the first of their compensation phases, given the timing and the effect of the CDK4/6 inhibitor. The R point should be mentioned in the Introduction and Discussion.

2) Despite the significance of the essential results, the work is not presented especially well. The Introduction fails to cover previous work accurately or comprehensively, the Results section is filled with distracting passages of discussion that are not results, and the descriptions of methods, analysis, and results are at several points cryptic and difficult to understand and evaluate. To convince reviewers that the conclusions are valid, and be accessible to eventual readers, the paper needs extensive revision.

3) The correlation of nuclear size and G1 length (Figure 1C) is rather poor, and doesn't support the authors' conclusion very strongly. Moreover, little evidence is provided supporting the authors' assumption that the relationship of nuclear size to cell size is fixed. Please provide such evidence if possible, to support the validity of using nuclear size measurement.

4) The experiments with drugs (Figures 5, 6) are very poorly presented and explained. I found myself incapable of interpreting these graphs, and wanting to see the raw data from these experiments. Why are there multiple points for each condition? Where is the color in Figure 5D? What are the units of "average growth rate" in Figure 5C-F and 6A? How can one calculate a cell cycle duration for a cell at three different time points during a single cell cycle (5E, F)? How do the plots support the conclusions in the text? I suggest that the authors revise their presentation of this data. However, even if it becomes more intelligible, the results obtained with the drugs have to be viewed as more preliminary than conclusive. A comprehensive analysis with one well characterized, specific, cell cycle inhibitor and one growth inhibitor could be more compelling.

5) The size compensation that is presented for the various cell cycle and growth inhibitors is convincing, and the fact that the Cdk4/6 inhibitor and Hsp90 inhibitor appear to override compensation is striking. The cell cycle compensation is validated genetically by cyclin E overexpression, but it would be useful to validate the Cdk4/6 inhibitor effect genetically as well, i.e., does an Rb phosphorylation site mutant line also fail in compensation? This would provide a compelling link between size control and one of the most frequently mutated axes in human cancer. See below for an additional point on Rb.

6) On the cyclin E experiment, I am concerned that the experiment seems to have been done by bulk transfection. This has two drawbacks, namely that cells will have been stressed by the transfection procedure and that protein levels may vary considerably from cell to cell. What was the transfection efficiency and how long were cells allowed to recover? A better experiment would have been to analyze 2 or 3 stable clones that express cyclin E under an inducible promoter, so perhaps the authors might consider this. As suggested above, a similar experiment could be done in parallel with non-phosphorylatable Rb or overexpressed CDK4/6 with the prediction that size homeostasis would be disrupted.

7) A strength of the manuscript is that similar results were obtained in highly malignant HeLa cells and normal RPE1 epithelial cells (which I assume were not immortalized with telomerase – is that true?). It is quite remarkable that HeLa cells have maintained size control despite the evidence for widespread aneuploidy, copy number variation, chromothripsis and disruption of cell cycle and growth regulatory pathways (PMID: 23550136). This deserves comment, particularly with respect to specific effects of HPV18 insertion, which should have disrupted the Rb axis (see above) and inactivated p53. It would be of some additional comfort in terms of the generality of the compensation mechanism if the authors could provide supporting data for a third cell line, for example an Rb- line. I am puzzled by the apparent intact compensation in HeLa cells but its disruption by the CDK4/6 inhibitor. Rb family member knockouts cause cells to be small, and might have been expected to be disrupted for one of the compensation points. Please comment on these points.

8) A key missing piece of information is the cell cycle stage at which size variance is actively diminished (e.g., in Figure 3G, H, J). The authors seem to be deliberately vague about this point, which should be clarified – when do the size compensation effects occur? Please state this clearly.

[Editors' note: further revisions were requested prior to acceptance, as described below.]

Thank you for submitting your article "Cell size sensing in animal cells coordinates anabolic growth rates and cell cycle progression to maintain cell size uniformity" to *eLife*. Your article has been evaluated by Marianne Bronner as the Senior Editor and reviewed by two peer reviewers, one of whom is a member of our Board of Reviewing Editors.

The reviewers have discussed the reviews with one another and the Reviewing Editor has drafted this decision to help you prepare a revised submission.

Both reviewers felt the paper was markedly improved, and were supportive of publication. Reviewer #1 felt that the exciting results you present with the Cdk4/6 and Hsp90 inhibitors should be validated by genetic loss-of-function tests, in order to rule out possible off-target, non-specific effects of Palbociclib and Radicicol. Alternatively, these observations might be validated with a second pair of Cdk4 and Hsp90 inhibitors. We agree that these validations are important, and so we'd like to encourage you to provide them, if they are technically feasible. On the other hand, if performing these controls is problematic we would appreciate receiving a justification for their absence. The reviews follow below for your reference. We look forward to receiving your revision.

*Reviewer #1:*

As in the original submission, this paper addresses of how cell size is controlled in tissue culture cells. This is a very basic question that has been debated for many years, with rather little significant progress towards a consensus understanding of the underlying mechanisms. Here, the problem is attacked logically, using highly accurate single cell methods. The paper comes to the conclusion that cell size is under active control during at least two phases in the division cycle in multiple cell types. In addition, the authors show that inhibition of either Cdk4 or Hsp90 can impair the mechanisms that reduce size variation, giving either very large cells or highly variable cells that are defective in coupling growth rates and size to cell division. These are significant findings. The revision of this interesting paper is markedly improved. It is shorter, more concise, and easier to understand. Many new measurements are presented, further supporting the authors' conclusions. However, the paper could still be clearer and more concise, and it is still the case that the amount of progress towards a mechanistic understanding of size control is modest. The paper is still essentially a characterization of a process, showing that cell size is actively adjusted during growth, and only superficial clues are given as to how this is accomplished, mechanistically. Moreover these clues, namely that the Cdk4/6 inhibitor Palbociclib and the Hsp90 inhibitor Radicocol dysregulate size control, and not followed any farther than before. At a minimum, I would expect genetic validations of these drug hits for the paper to be published. This is important to support the conclusions. Specifically, the authors should assess size control in cells lacking Cdk4/6 and Hsp90 gene function, using gene knock-outs by siRNA or deletion. In addition, I would encourage the authors to strive further for a shorter, more readable manuscript.

*Reviewer #2:*

Overall, the manuscript has been improved considerably and is a much more compelling read. The main points raised in the previous reviews have been very nicely addressed by new experiments of high quality, particularly the inducible experiments with cyclin D, cyclin E and p27 and the additional cell line experiments with cells of different Rb status. Moreover, the Introduction and Discussion now frame the size control problem and the results of the study properly. I have no major concerns and believe the manuscript should be quickly published on the basis of the key main finding of an active feedback mechanism in size control, which is likely to become a landmark discovery in the size control field.

---

## [Author Response]

Essential revisions (from the reviews):1) The Introduction needs revision.(from review #1): The Introduction fails to outline the "sizer" models proposed, and experimentally supported for budding and fission yeast. These should be explained, as they are the best examples of size control in the literature. In addition I would disagree with the authors statement that the field believes that "growth and cell cycle progression are not coordinated." On the contrary, there's massive evidence showing that cell growth regulates cell cycle progression. Please re-write the intro to better reflect the literature and beliefs in the field.(from review #2): The authors could do a better job framing the size control problem, perhaps starting by describing the observations of Zetterberg et al. briefly. I don't agree with the claim that the consensus view is that size control doesn't exist in animal cells. The weight of evidence for size control going all the way back to Zetterberg is strong, and the review cited to the contrary is an outlier opinion that isn't by any means representative, and thus I don't think it is appropriate to quote from this review. The fixation on the general issue of active size control to some extent distracts from the key finding that there is a bona fide size-dependent compensation mechanism, as opposed to a simple threshold. On this note, the authors make no mention all of the restriction point, which would seem a logical candidate for the first of their compensation phases, given the timing and the effect of the CDK4/6 inhibitor. The R point should be mentioned in the Introduction and Discussion.

Following the reviews suggestion, we have completely rewritten the Introduction. In the new Introduction, we cover literature on cell size in both animal cells and in yeast. Other subjects covered in the new Introduction include the distinction between the *sizer model* and the *adder model*, the titration/dilution model of size sensing (Whi5,Cln3), geometric models of size sensing (Cdr2/Pom1), and Zetterberg’s original publications.

2) Despite the significance of the essential results, the work is not presented especially well. The Introduction fails to cover previous work accurately or comprehensively, the Results section is filled with distracting passages of discussion that are not results, and the descriptions of methods, analysis, and results are at several points cryptic and difficult to understand and evaluate. To convince reviewers that the conclusions are valid, and be accessible to eventual readers, the paper needs extensive revision.

In the revised version, we thoroughly edited the Results section to increase clarity. In addition, certain sections were removed from the Results section and, after significant re-editing, were transferred to the Discussion.

3) The correlation of nuclear size and G1 length (Figure 1C) is rather poor, and doesn't support the authors' conclusion very strongly. Moreover, little evidence is provided supporting the authors' assumption that the relationship of nuclear size to cell size is fixed. Please provide such evidence if possible, to support the validity of using nuclear size measurement.

We address this concern with four separate approaches:

1) We provide new measurements to validate use of nuclear area as a proxy of cell size. These new measurements are included in Figure 2 and Figure 2—figure supplement 2 of the revised manuscript. These new measurements are discussed in the Results section of the revised manuscript, in a section that begins with the sentence “In yeast, the nucleus is known to grow continuously throughout all stages of cell cycle and is correlated with cell size (Jorgensen et al., 2007). To test whether this is also the case with our experimental system, we measured the correlation between nucleus size and cell size […]”

Briefly, to show that nuclear area is a reliable proxy for cell size we present the following measurements:

A) We show that the correlation of cell size and nucleus size is very significant (Figure 2A).

B) We show that both cell size and nucleus size steadily increase as cells progress in cell cycle (Figure 2B-2C and Figure 2—figure supplement 2). Also, these figures show that the increase in nucleus size and the increase in cell size are linearly correlated. Figure 2C also shows that the resolution of the nucleus area measurements is sufficient to discriminate average size differences that result from less than three hours of cell growth (~15% of cell cycle length).

C) In Figure 2—figure supplement 2, we show that nuclear growth is not restricted to S phase and is continuous throughout the whole of cell cycle. In fact, to show that growth in nuclear size does not reflect DNA replication, we show that the size of the nucleus steadily increases in cells that are arrested with aphidicolin (a DNA replication inhibitor).

2) We validated that, while the correlation of nucleus size and G1 length is weak, it is *significant* and highly *reproducible*. Measurements on the correlation of nucleus size versus G1 length were independently repeated at least six times, with very consistent results. In fact, a repeat of the same measurement also appears in Liu et al., a manuscript that was co-submitted with this one and received very positive reviews in *eLife*.

3) To the new Discussion, we added a section that specifically address the reasons that explain why this correlation is not strong. This new section begins with the sentence “While the correlation of cell cycle length and nucleus size is statistically significant (p<6.7x10^-8^) and reproducible (N>4), there remains the question of why this correlation is small in magnitude […]”

4) The *purpose* of the measurement on the correlation of G1 length and nuclear area is to support the claim that smaller cells spend longer periods of time in G1. To that end, it is worth noting that the correlation of nuclear size and G1 length is only one of several pieces of evidence that we provide in that direction. In fact, excluding Figure 4, all other figures in the manuscript include evidence in that direction. Specifically, in Figure 1D we show difference in the average cell size of G1 cells versus S phase cells that are not explained by age. Also, in Figure 3 and Figures 6-10, we show that the length of G1 is increased by pharmacological and genetic perturbations that decrease cell size.

4) The experiments with drugs (Figures 5, 6) are very poorly presented and explained. I found myself incapable of interpreting these graphs, and wanting to see the raw data from these experiments. Why are there multiple points for each condition? Where is the color in Figure 5D? What are the units of "average growth rate" in Figure 5C-F and 6A? How can one calculate a cell cycle duration for a cell at three different time points during a single cell cycle (5E, F)? How do the plots support the conclusions in the text? I suggest that the authors revise their presentation of this data. However, even if it becomes more intelligible, the results obtained with the drugs have to be viewed as more preliminary than conclusive. A comprehensive analysis with one well characterized, specific, cell cycle inhibitor and one growth inhibitor could be more compelling.

In Figure 6, of the revised manuscript, we show new measurements from a ‘comprehensive analysis with one well characterized cell cycle inhibitor and one growth inhibitor’. Specifically, we performed measurements of cell size and cell count over time from populations that were treated with SNS032 (a well characterized cell cycle inhibitor) and with rapamycin (a well characterized growth inhibitor). We use these measurements to demonstrate the coordination of growth rate and cell size. Specifically, that we use these two drugs to show that perturbations of growth rate result in compensatory increases in cell cycle length and, vice versa, that perturbations of cell cycle length result in compensatory changes of growth rate.

In addition, to increase the clarity on this experiment, we added new figures (Figures 7,8) that presents the raw data. Last, we revised the text to significantly increase clarity of the description of this experiment.

5) The size compensation that is presented for the various cell cycle and growth inhibitors is convincing, and the fact that the Cdk4/6 inhibitor and Hsp90 inhibitor appear to override compensation is striking. The cell cycle compensation is validated genetically by cyclin E overexpression, but it would be useful to validate the Cdk4/6 inhibitor effect genetically as well, i.e., does an Rb phosphorylation site mutant line also fail in compensation? This would provide a compelling link between size control and one of the most frequently mutated axes in human cancer. See below for an additional point on Rb.6) On the cyclin E experiment, I am concerned that the experiment seems to have been done by bulk transfection. This has two drawbacks, namely that cells will have been stressed by the transfection procedure and that protein levels may vary considerably from cell to cell. What was the transfection efficiency and how long were cells allowed to recover? A better experiment would have been to analyze 2 or 3 stable clones that express cyclin E under an inducible promoter, so perhaps the authors might consider this. As suggested above, a similar experiment could be done in parallel with non-phosphorylatable Rb or overexpressed CDK4/6 with the prediction that size homeostasis would be disrupted.

Figure 10, in the revised manuscript, shows measurements on three different stable cell lines that we generated to comply with these reviewer comments:

1) A cell line with doxycycline-inducible expression of cyclin E

2) A cell line with doxycycline-inducible expression of cyclin D

3) A cell line with doxycycline-inducible expression of p27

As expected, doxycycline dependent over expression of cyclin E and cyclin D result in increased rates of cell division (shorter cell cycles) while overexpression of p27 results in slower rates of cell division (longer cell cycles). In perfect consistency with the pharmacological measurements, the decreased cell cycle length that is caused by cyclin E overexpression is compensated by an increased growth rate such that cell size remains unchanged. In contrast, when cell cycle length is shortened by overexpression of cyclin D, cell size is significantly reduced.

7) A strength of the manuscript is that similar results were obtained in highly malignant HeLa cells and normal RPE1 epithelial cells (which I assume were not immortalized with telomerase – is that true?). It is quite remarkable that HeLa cells have maintained size control despite the evidence for widespread aneuploidy, copy number variation, chromothripsis and disruption of cell cycle and growth regulatory pathways (PMID: 23550136). This deserves comment, particularly with respect to specific effects of HPV18 insertion, which should have disrupted the Rb axis (see above) and inactivated p53. It would be of some additional comfort in terms of the generality of the compensation mechanism if the authors could provide supporting data for a third cell line, for example an Rb- line. I am puzzled by the apparent intact compensation in HeLa cells but its disruption by the CDK4/6 inhibitor. Rb family member knockouts cause cells to be small, and might have been expected to be disrupted for one of the compensation points. Please comment on these points.

In the original manuscript, conclusions were derived from measurements on two cell lines, hTERT-immortalized Rpe1 cells and HeLa cells. To the revised manuscript, we now added measurements from three additional cell lines: two cell lines that lack Rb activity (Rb-null SAOS2 cells and SV40-immortalized 16HBE cells) and one cell line that has intact Rb signaling (U2OS). Measurements on all cell lines are shown in Figure 9 of the revised manuscript. The raw data from measurements on the new cell lines is shown in supplements to Figure 9.

Consistent with what was shown in the original manuscript, the coordination of growth rate and cell cycle length is observed also in the cell lines that lack Rb activity. While Figures 8 and 10 show that genetic and chemical perturbations of cyclin D / Cdk4 cause significant changes in cell size, Figure 9 shows that Rb is not necessary for the coordination of growth rate and cell cycle length. In the Discussion, we devote a section to consider the implications of these results. We hypothesize that the Rb/CDK4 axis may have roles in determining target size but not in size sensing and suggest this possibility as grounds for future research.

An additional note regarding the following comment:

"I am puzzled by the apparent intact compensation in HeLa cells but its disruption by the Cdk4/6 inhibitor. Rb family member knockouts cause cells to be small, and might have been expected to be disrupted for one of the compensation points. Please comment on these points."

In the original study, while the coordination of growth rate and cell cycle length was seen in both Rpe1 and HeLa, its disruption upon Cdk4/6 inhibition with palbociclib was only seen in Rpe1. HeLa cells were insensitive to palbociclib, as expected in an Rb-inactive cell line. This was unclear in the original manuscript and is made explicitly clear in the revised manuscript by presenting the results from both cell lines side by side in Figure 9 (along with results from the additional cell lines discussed above).

8) A key missing piece of information is the cell cycle stage at which size variance is actively diminished (e.g., in Figure 3G, H, J). The authors seem to be deliberately vague about this point, which should be clarified – when do the size compensation effects occur? Please state this clearly.

The Gcv-value analysis consistently identifies two periods of decreasing cell size variance. One of these two periods coincides in time with the average time of the G1/S transition and another coincides in time with late S-phase/G2. As the reviewer’s question implies, these data suggest that the process causing the decreased variance is regulated by components of the cell cycle clock. However, we thought that drawing this conclusion in our manuscript would be premature. Linking these drops in variance with any of the molecular processes that define the cell cycle (like cyclin degradation or DNA replication) is not yet possible with the current experimental design, due to several concerns. Firstly, we note that the observed reduction in cell size variance is the end-product of a process that occurred earlier in time. Specifically, it is the cumulative effect of size-dependent growth rate regulation that will lead to an eventual decrease in cell size variance, so we do not want to over-interpret the precise timing of the variance drop before further experiments are done.

Furthermore, it is possible that the observed periods of active cell size correction are not actually triggered by specific cell cycle events occurring during those periods, but instead coincide with them due to a confounding factor. For example, the dip in cell size variance (and the corresponding dip in size/growth rate correlation observed in single-cell growth trajectories) that coincides with the average time of S-phase entry may reflect a size-corrective mechanism that is activated at that time. However, it is also possible that this corrective mechanism is actually triggered by cells reaching a particular size (e.g. cells above a certain size slow their growth), and that, since most cells are born small, the average cell only comes into the range of this corrective mechanism towards the end of G1. Because cell cycle progression and cell size are highly correlated, it is difficult to determine whether the growth rate regulation we observed is directly triggered by S-phase entry.

The fact that we observe two distinct periods of size-dependent growth rate regulation during the cell cycle is significant and does indicate a cell-cycle-dependent change in the mode of size regulation. (Even if regulation is directly triggered by size, as in the example above, the presence of two dips would suggest that the “target size” is raised for cells later in the cell cycle.) However, we cannot assume that the molecular events responsible for this change precisely coincide with the observed dips in cell size variance.

We are currently pursuing several approaches to link the drops in cell size variance with specific cell cycle regulators. We have expanded the assays described in this manuscript and optimized them for use in combination with high-throughput perturbation screens. However, as it will take much more time before we can obtain significant results, we hope to resolve this question in a future publication.

[Editors' note: further revisions were requested prior to acceptance, as described below.]

Reviewer #1:As in the original submission, this paper addresses of how cell size is controlled in tissue culture cells. This is a very basic question that has been debated for many years, with rather little significant progress towards a consensus understanding of the underlying mechanisms. Here, the problem is attacked logically, using highly accurate single cell methods. The paper comes to the conclusion that cell size is under active control during at least two phases in the division cycle in multiple cell types. In addition, the authors show that inhibition of either Cdk4 or Hsp90 can impair the mechanisms that reduce size variation, giving either very large cells or highly variable cells that are defective in coupling growth rates and size to cell division. These are significant findings. The revision of this interesting paper is markedly improved. It is shorter, more concise, and easier to understand. Many new measurements are presented, further supporting the authors' conclusions. However, the paper could still be clearer and more concise, and it is still the case that the amount of progress towards a mechanistic understanding of size control is modest. The paper is still essentially a characterization of a process, showing that cell size is actively adjusted during growth, and only superficial clues are given as to how this is accomplished, mechanistically. Moreover these clues, namely that the Cdk4/6 inhibitor Palbociclib and the Hsp90 inhibitor Radicocol dysregulate size control, and not followed any farther than before. At a minimum, I would expect genetic validations of these drug hits for the paper to be published. This is important to support the conclusions. Specifically, the authors should assess size control in cells lacking Cdk4/6 and Hsp90 gene function, using gene knock-outs by siRNA or deletion. In addition, I would encourage the authors to strive further for a shorter, more readable manuscript.

The referee raises three points:

A) Writing style and clarity can be improved.

We accept the criticism and have improved clarity in the revised manuscript.

B) The study will benefit from genetic validation of the influence of CDK4/6 and HSP90 on cell size.

We assessed size control in cells lacking cdk4/6 using siRNA but left validation of HSP90 for future work. In the original manuscript, we reported evidence suggesting an influence of CDK4/6 activity on cell size. In our original manuscript, evidence linking CDK 4/6 activity with cell size derived from two sources; pharmacological *inhibition* of CDK 4/6 activity and genetic *activation* of CDK 4/6 activity. Pharmacological inhibition of CDK 4/6 resulted in a marked increase of cell size. To further validate the influence of CDK 4/6 on cell growth we performed a second experiment whereby CDK4 activity was activated rather than chemically inhibited. Our hypothesis was that, since inhibition of CDK4/6 results in larger cells, the activation of CDK4/6 should result in smaller cell size. To activate CDK4/6 we used an RPE1 cell line with doxycycline dependent expression of the CDK 4/6 activator, cyclin D. As we anticipated, cyclin D overexpression resulted in a significant reduction in cell size. In this final round of reviewer comments, the referee suggested a final control: knock down CDK 4/6 and test whether the results replicate the increased cell size observed with chemical inhibition of CDK activity. Conforming to the reviewer suggestion, we used siRNA to knock down CDK4, CDK6 or both. In all cases, knockdown of CDK 4/6 resulted in increased cell size (Figure 8—figure supplement 1), replicating the result obtained with the chemical inhibitor, palbociclib. To conclude, we now have three lines of evidence implicating CDK 4/6 activity with increased cell size phenotype. One, pharmacological inhibition of CDK 4/6 causes larger cells. Two, genetic activation of CDK 4/6 with overexpression of cyclin D causes smaller cells. Third, genetic inhibition of CDK 4/6 with siRNA causes larger cells. We believe that this substantially grounds an influence of CDK 4/6 on cell size. A question that still remains open is: How does CDK 4/6 influence cell size? Since the roles of CDK 4/6 in size regulation are not a main theme in our study, we choose to leave this question for a later study.

C) The referee correctly asserts that progress towards a mechanistic understanding of size control is modest.

We agree, yet with a reservation. The current paper is part of a larger study. Mechanisms for G1 length regulation are described in the accompanying manuscript (Liu et al.). Also, more generally, mechanisms can be sought only once phenomenon have been described. In this paper we describe phenomenon that have not yet been demonstrated.